Research

 

**Subject Area:**
cellular biology/molecular biology

osteocyte, TSC1, sclerostin, mTORC1, Sirt1, osteogenesis

**Authors for correspondence:**
Zhong-Kai Cui
e-mail: zhongkaicui@smu.edu.cn
Anling Liu
e-mail: aliu@smu.edu.cn
Xiaochun Bai
e-mail: baixc15@smu.edu.cn

[†]These authors contributed equally to this work.

# Osteocyte TSC1 promotes sclerostin secretion to restrain osteogenesis in mice

Wen Liu[1,†], Zhenyu Wang[2,†], Jun Yang[1], Yongkui Wang[1], Kai Li[1,2], Bin Huang[2], Bo Yan[2], Ting Wang[1], Mangmang Li[1], Zhipeng Zou[1], Jian Yang[1,4], Guozhi Xiao[5], Zhong-Kai Cui[1], Anling Liu[3] and Xiaochun Bai[1]

[1]Key Laboratory of Mental Health of the Ministry of Education, Department of Cell Biology, School of Basic Medical Science, [2]Academy of Orthopedics, Guangdong Province, Guangdong Provincial Key Laboratory of Bone and Joint Degenerative Diseases, The Third Affiliated Hospital, and [3]Department of Biochemistry and Molecular Biology, School of Basic Medical Science, Southern Medical University, Guangzhou, People's Republic of China
[4]Department of Biomedical Engineering, Materials Research Institute, The Huck Institutes of the Life Sciences, The Pennsylvania State University, University Park, PA, USA
[5]Department of Biochemistry and Department of Biology and Shenzhen Key Laboratory of Cell Microenvironment, South University of Science and Technology of China, Shenzhen, People's Republic of China

(ID) Z-KC, 0000-0003-3112-9379; AL, 0000-0001-9664-0245; XB, 0000-0001-9631-4781

Osteocytes secrete the glycoprotein sclerostin to inhibit bone formation by osteoblasts, but how sclerostin production is regulated in osteocytes remains unclear. Here, we show that tuberous sclerosis complex 1 (TSC1) in osteocytes promotes sclerostin secretion through inhibition of mechanistic target of rapamycin complex 1 (mTORC1) and downregulation of Sirt1. We generated mice with *DMP1*-Cre-directed *Tsc1* gene deletion (*Tsc1* CKO) to constitutively activate mTORC1 in osteocytes. Although osteocyte TSC1 disruption increased RANKL expression and osteoclast formation, it markedly reduced sclerostin production in bone, resulting in severe osteosclerosis with enhanced bone formation in mice. Knockdown of TSC1 activated mTORC1 and decreased sclerostin, while rapamycin inhibited mTORC1 and increased sclerostin mRNA and protein expression levels in MLO-Y4 osteocyte-like cells. Furthermore, mechanical loading activated mTORC1 and prevented sclerostin expression in osteocytes. Mechanistically, TSC1 promotes sclerostin production and prevents osteogenesis through inhibition of mTORC1 and downregulation of Sirt1, a repressor of the sclerostin gene *Sost*. Our findings reveal a role of TSC1/mTORC1 signalling in the regulation of osteocyte sclerostin secretion and bone formation in response to mechanical loading *in vitro*. Targeting TSC1 represents a potential strategy to increase osteogenesis and prevent bone loss-related diseases.

## 1. Introduction

Osteocytes are the most abundant osteolineage cells in bone and mechanical stress sensor cells which embedded in the cortical and cancellous bone matrix [1,2]. Osteocytes are descended from mesenchymal stem cells through osteoblastic differentiation [3–5]. Over the past several decades, the role of osteocytes as the professional mechano-sensory cells of bone has been well established [6,7]. Recently, accumulating evidence has shown that osteocytes play a central role in coordinating the function of osteoblasts and osteoclasts through cell-to-cell communication. Osteocytes react to mechanical loading by maintaining and monitoring bone remodelling and regulating the homeostasis between osteoblasts and osteoclasts via secretion of cytokines [8–11]. Therefore, understanding the biology of osteocyte function is essential for the prevention and treatment of metabolic diseases of bone.

Sclerostin, encoded by the *Sost* gene, is a glycoprotein secreted by osteocytes [12]. Sclerostin is a strong negative regulator of osteoblast differentiation and bone formation, acting in a paracrine fashion, partially in response to mechanical loading [13]. By competitively binding to the WNT co-receptor, namely low-density lipoprotein receptor-related protein 5/6 (LRP5/6), previous studies have demonstrated that sclerostin induces phosphorylation and degradation of β-catenin and hinders the activation of osteoblasts [14,15]. In addition, the expression of sclerostin was also regulated by HDAC5 and PTH [16,17]. Mutations or deletions 52 kb downstream in the *Sost* gene locus are associated with two rare autosomal recessive disorders, sclerosteosis and van Buchem disease [18,19], characterized by excessive bone growth [20]. Transgenic mice overexpressing SOST exhibit low bone mass [21], whereas targeted deletion of the *Sost* gene in mice results in increased formation of bone and improves healing of bone defects [22]. Treatment with anti-sclerostin monoclonal antibodies inhibited the activity of sclerostin, thus improving bone mass and bone strength, along with enhancing repair of fractures in mice and rats [23,24]. An antibody that targets sclerostin (romosozumab) has already passed phase III clinical trials for the treatment of osteoporosis and is expected to become a new therapeutic [25]. Although it is established that osteocytes coordinate the osteogenic response to mechanical force by downregulating sclerostin, thereby locally unleashing Wnt signalling in osteoblasts, the mechanisms that regulate sclerostin expression in osteocytes are not well defined.

The tumour suppressor genes *TSC1* and *TSC2*, respectively, encode hamartin and tuberin, which are involved in direct control of the activity of mechanistic target of rapamycin complex 1 (mTORC1). TSC1/2 complex expresses GTPase activating protein bioactivity towards small GTPase Rheb, thereby inducing conversion of active GTP-bound Rheb to inactive GDP-bound Rheb. Active Rheb promotes mTORC1 activation, controlling protein translation by phosphorylating the ribosomal protein S6 kinase (S6K1) and controlling cap-dependent protein translation [26]. mTORC1 abnormal activation induced by mutations of TSC1 or TSC2 contributes to the development of the genetic disorder tuberous sclerosis complex (TSC), which is defined by benign hamartomas in several organ systems [27]. The mechanism by which TSC acts in the skeletal system, causing effects such as sclerotic bone lesions, is less well understood. Although the roles of TSC/mTORC1 have been extensively demonstrated in cell metabolism and metabolic diseases [28], their functions and regulatory mechanisms in bone metabolism are not fully investigated. We and other groups have recently shown that TSC1/2 plays a critical role in the proliferation, differentiation and function of bone marrow-derived stroma cells (BMSCs), osteoclasts and osteoblasts [29–32]. However, the specific role of TSC1/mTORC1 in mature osteocytes is unknown.

In this study, we generated osteocyte *Tsc1* deletion mice (*Tsc1* CKO) to constitutively activate mTORC1 in osteocytes. We found that osteocyte TSC1 disruption reduced sclerostin in bone, leading to osteosclerosis with enhanced bone formation in mice. TSC1 promotes sclerostin expression and prevents osteogenesis through inhibition of mTORC1 and downregulation of Sirt1. Our findings reveal an important role of TSC1/mTORC1 signalling in the regulation of osteocyte sclerostin secretion and bone formation in response to mechanical loading *in vitro*. Targeting TSC1 may present a

strategy to increase osteogenesis and prevent bone loss-related diseases.

## 2. Result

### 2.1. Deletion of TSC1 in osteocytes markedly increases bone mass in mice

To characterize the functional role of TSC1 in osteocytes, we generated mice in which TSC1 was deleted in osteocytes. We mated *Tsc1*-flox mice with *DMP1*-Cre mice to generate a mouse model with conditional *Tsc1* knockout mice (*DMP1*-Cre$^+$; *Tsc1*$^{f/f}$/*Tsc1* CKO) and littermate controls (*DMP1*-Cre$^−$; *Tsc1*$^{f/f}$/DTCL) for detailed analysis (electronic supplementary material, figure S1A and B). Furthermore, the DMP1-mediated excision was confirmed by a double-fluorescent Cre reporter mouse (mT/mG), in which osteocytes showed red fluorescence in mG complementary to mT and the data verified DMP1-mediated excision mainly in osteocytes (electronic supplementary material, figure S1C). *Tsc1* CKO mice were born at the expected Mendelian frequency. Both western blotting and immunohistochemical staining of distal femur sections showed a massive increase in phosphorylation of mTORC1 downstream substrate S6 (Ser235/236) in osteocytes of *Tsc1* CKO mice (figure 1a,b), indicating that mTORC1 was activated by *Tsc1* disruption. At the gross level, there was no significant difference in the body weight or length between *Tsc1* CKO mice and their control littermates (electronic supplementary material, figure S2A and B), and the representative images of femur, tibia and spinal column of *Tsc1* CKO mice did not significantly differ from DTCL mice (figure 1c). Micro-computed tomography (micro-CT) analysis of 4-, 8- and 12-week-old mice revealed an increased bone mass in *Tsc1* CKO mice. The distal regions of the femur and the fifth lumbar vertebra exhibited a marked increase in cancellous bone mass in *Tsc1* CKO mice compared with that in DTCL, as reflected in bone mineral density (BMD), trabecular thickness (Tb. Th) and trabecular number (Tb. N), bone volume/tissue volume (BV/TV), as well as a slight decrease in trabecular separation (Tb. Sp) (figure 1d; electronic supplementary material, figure S2C and D and tables S1–S3). Analysis of the femoral mid-shaft revealed a significant increase in cortical bone mass in *Tsc1* CKO mice (figure 1e,f), while BMD was not significantly different from that of the littermate controls (electronic supplementary material, tables S1–S3). Haematoxylin and eosin (H&E) staining of long bones indicated a drastic increase in bone mass in *Tsc1* CKO mice (figure 1g). Altogether, these data suggest that TSC1 deletion and mTORC1 activation in osteocytes markedly increase bone mass in mice.

### 2.2. Deletion of TSC1 in osteocytes stimulates osteogenesis and bone formation in mice

Consistent with the high bone mass phenotype in *Tsc1* CKO mice, bone mass on the endocortical surface was notably increased in cancellous bone of femur in these mutant mice (figure 2a,b). Histomorphological analysis showed that the thickness of cortical bone and the expression of osteocyte marker DMP1 were also increased in *Tsc1* CKO mice (electronic supplementary material, figure S3A; figure 2c,d). To

royalsocietypublishing.org/journal/rsob Open Biol. 9: 180262

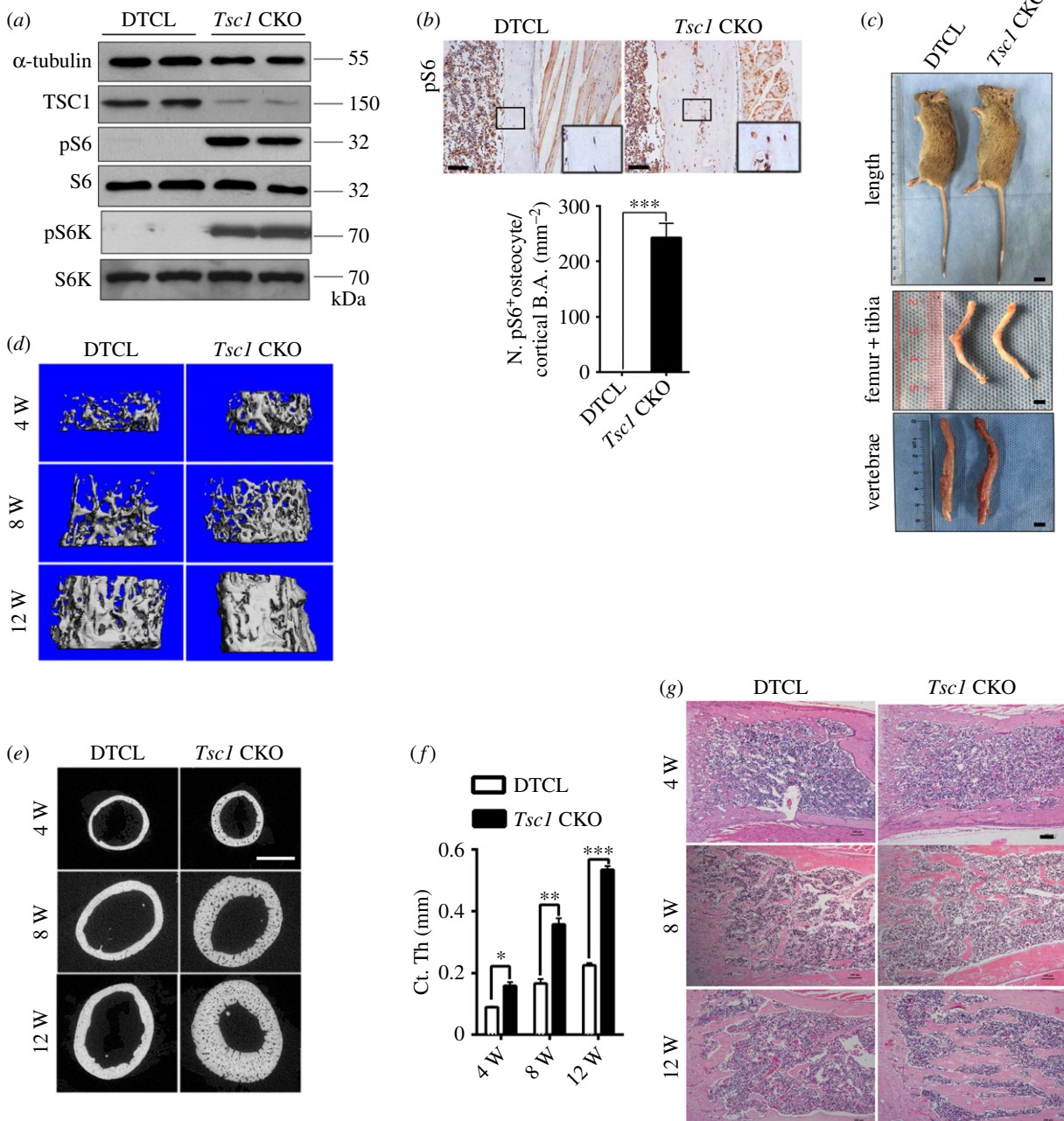

**Figure 1.** Deletion of TSC1 in osteocytes markedly increased bone mass in mice. (a) Western blot analysis of pS6 (Ser235/236), pS6K and, TSC1 in femora of 10-week-old control (DTCL) and *Tsc1* CKO mice. pS6, phospho-S6, pS6K, phospho-S6 K. (b) Immunohistochemical staining of pS6 (Ser235/236) in distal femora of 10-week-old *Tsc1* CKO mice. The boxed area is magnified below. Scale bars, 100 μm. Quantification (below) shows the pS6-positive cell numbers (N. pS6$^+$) between DTCL and *Tsc1* CKO mice (*t*-test, $p < 0.0001$, $n = 6$). (c) Images of 10-week-old *Tsc1* CKO mice, length, femur and tibia, vertebral column. No difference in *Tsc1* CKO mice was detected when compared with control mice. The scale bars represent 1 cm ($n = 6$). Data are represented as mean $\pm$ s.d., and ***$p < 0.001$. (d) Representative images of micro-CT analyses of the structure of metaphyseal trabecular bone and cortical bone in the distal femora in 4-, 8- and 12-week-old control (DTCL) and *Tsc1* CKO mice ($n = 6$). (e) Representative micro-CT two-dimensional images of cortical bone of 4-, 8- and 12-week-old control and *Tsc1* CKO mice. Scale bar, 1 mm ($n = 6$). (f) Cortical bone mass in the distal femora in 4-, 8- and 12-week-old control and *Tsc1* CKO mice (*t*-test, 4 weeks: $p = 0.0475$, 8 weeks: $p = 0.0078$, 12 weeks: $p = 0.0001$, $n = 6$). (g) H&E staining of long bone from 4-, 8- and 12-week-old *Tsc1* CKO and control mice. The scale bars represent 100 μm ($n = 6$).

further investigate osteoblastic bone formation activity in *Tsc1* CKO mice, we performed double fluorochrome labelling analyses. Incorporation of the two fluorochromes was evident in the bones of control mice.

Although the mineralizing surface was dramatically and abnormally increased in cortical bone, the distance between calcein-labelled mineralization fronts at the endosteum of the mid-shaft of the femur was greater in *Tsc1* CKO mice than in controls (figure 2e). Histomorphometric

measurements showed that the endosteum mineral apposition rate (MAR), mineralizing surface/bone surface (MS/BS) and bone formation rate (BFR) of control mice were lower than those of *Tsc1* CKO mice (figure 2f; electronic supplementary material, figure S3B and C), suggesting increased bone formation in *Tsc1* CKO mice. In addition, osteocytes lacking *Tsc1* exhibited activation of mTORC1 in osteocytes and enhanced osteocalcin (OCN) expression in osteoblasts (electronic supplementary material, figure S3D; figure 2g,h).

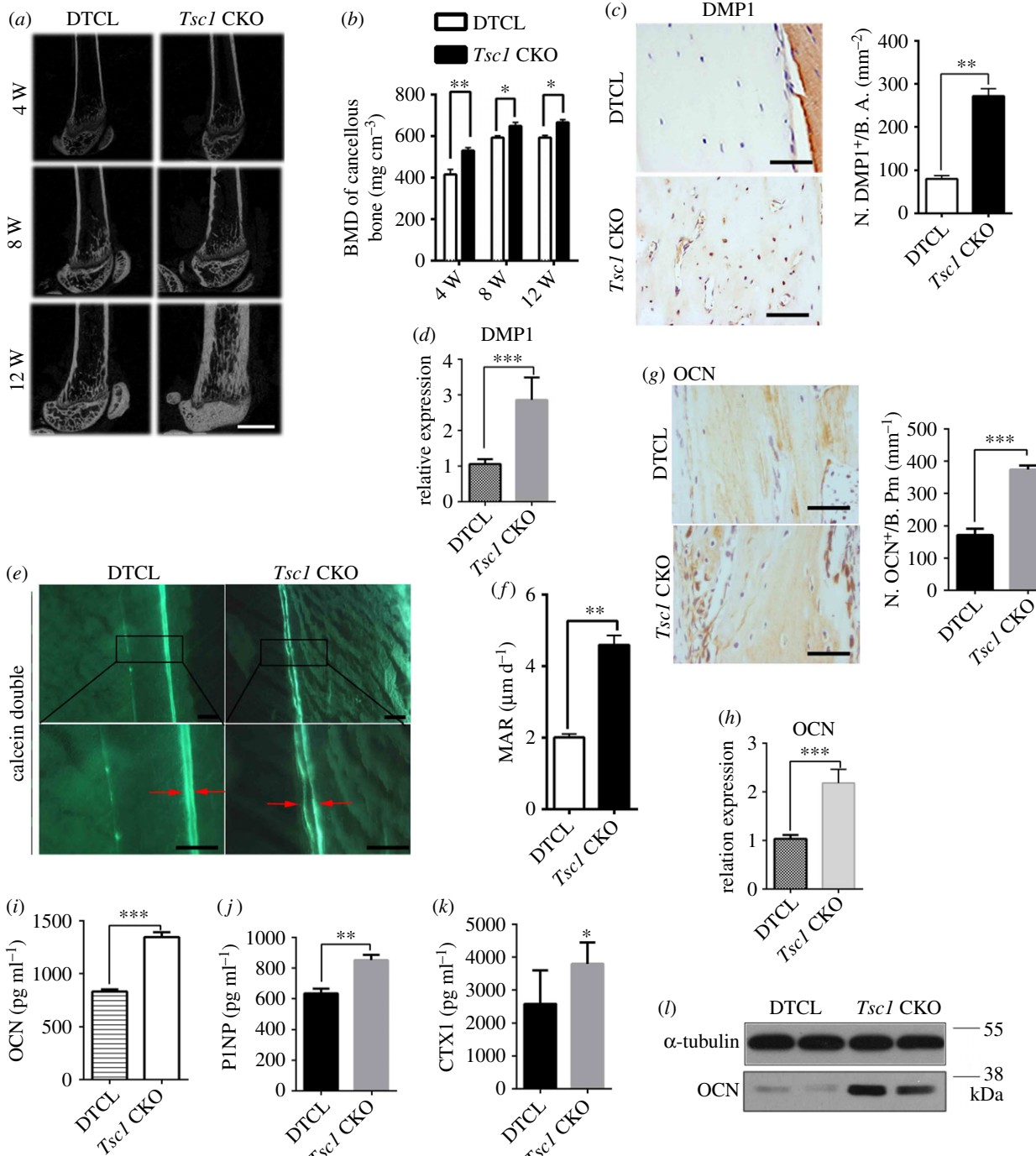

**Figure 2.** Deletion of TSC1 in osteocytes stimulates osteogenesis and bone formation in mice. (a) Micro-CT reconstruction of distal femora from 4-, 8- and 12-week-old control (DTCL) and *Tsc1* CKO mice. The scale bar represents 1 mm ($n = 6$). (b) BMD of trabecular bone in the distal femora from 4-, 8- and 12-week-old control (DTCL) and *Tsc1* CKO mice (t-test, 4 weeks: $p = 0.0083$, 8 weeks: $p = 0.0257$, 12 weeks: $p = 0.0284$). BMD, bone mineral density ($n = 6$). (c) Immunohisto-chemical staining and quantification of DMP1 in decalcified sections of distal femora from 10-week-old control (DTCL) and *Tsc1* CKO mice (t-test, $p = 0.0037$). Numbers of DMP1 (N. DMP1$^+$). The scale bars represent 50 μm ($n = 6$). (d) mRNA expression of DMP1 in femora from DTCL and *Tsc1* CKO mice (t-test, $p = 0.0002$, $n = 6$). (e) Calcein double labelling of cortical bone in distal femora from 10-week-old control (DTCL) and *Tsc1* CKO mice. The boxed area is magnified in the panel below. The scale bars represent 100 μm and 50 μm ($n = 6$). (f) Mineral apposition rate (MAR) (t-test, $p = 0.0062$). (g) Immunohistochemical staining and quantification analysis of OCN in distal femora from 10-week-old control and *Tsc1* CKO mice (t-test, $p = 0.0008$). Numbers of OCN (N. OCN$^+$). The scale bars represent 50 μm ($n = 6$). (h) mRNA level of osteocalcin (OCN) in DTCL and *Tsc1* CKO mice (t-test, $p = 0.0008$). (i) Enzyme-linked immunosorbent assay (ELISA) detection of serum OCN levels in control (DTCL) and *Tsc1* CKO mice (t-test, $p = 0.0009$, 12-week-old males, $n = 6$). (j) ELISA detection of serum P1NP levels in control and *Tsc1* CKO mice (t-test, $p = 0.001$, 12-week-old mice, $n = 6$). (k) ELISA detection of serum CTX-1 levels in control and *Tsc1* CKO mice (t-test, $p = 0.029$, 12-week-old mice, $n = 6$). (l) Western blot analysis of the expression levels of OCN protein in trabecular bone lysates from control and *Tsc1* CKO mice. All experiments were repeated independently three times. Data are represented as mean $\pm$ s.d., $*p < 0.05$, $**p < 0.01$ and $***p < 0.001$.

The serum levels of OCN and procollagen 1 N-terminal propeptide (P1NP), a serum bone formation marker, and bone resorption marker CTX-1 were also enhanced in *Tsc1* CKO mice compared with those of control mice

(figure 2i–k). Western blot analysis further confirmed that the expression of OCN was drastically increased in bone lysates of *Tsc1* CKO mice compared with those of DTCL mice (figure 2l). Altogether, these data suggest that TSC1

royalsocietypublishing.org/journal/rsob Open Biol. 9: 180262

deletion and mTORC1 activation in osteocytes markedly increased bone mass in mice.

## 2.3. Deletion of TSC1 in osteocytes stimulates RANKL expression and osteoclast formation

Evidence has shown that osteocytes are the major source of RANKL in bone to stimulate osteoclast formation [33]. We next examine whether TSC1 regulates RANKL expression in osteocytes and osteoclast formation in bone. Tartrate-resistant acid phosphatase (TRAP) staining revealed that the osteoclast number was significantly increased in the cancellous bone and endocortical surface of *Tsc1* CKO mice (figure 3*a*,*b*). Western blot analysis of bone lysates showed increased RANKL in *Tsc1* CKO mice (figure 3*c*). To further determine the role of osteocyte TSC1/mTORC1 in osteoclastogenesis, we transfected osteocyte-like cell line MLO-Y4 with lentivirus carrying TSC1 shRNA and/or treated MLO-Y4 with 1 nM rapamycin to activate or inhibit mTORC1 activity. The activity of TSC1 shRNA transfection was verified by green fluorescent protein (GFP) fluorescence and western blot assay of pS6 and pS6K (figure 3*d*,*e*). We found that downregulation of TSC1 by shTSC1 virus increased the protein and mRNA expression levels of RANKL and decreased the protein and mRNA expression levels of osteoprotegerin (OPG) (figure 3*e*–*g*), while rapamycin treatment inhibited RANKL protein and mRNA expression and enhanced OPG protein and mRNA expression (figure 3*h*–*j*). Taken together, these data suggest that TSC1 regulates RANKL expression in osteocytes, thus affecting osteoclast formation *in vitro* and in mice.

## 2.4. TSC1 inhibits sclerostin expression in osteocytes in an mTORC1-dependent manner

We next investigate the mechanism by which osteocyte TSC1 regulates bone formation. Sclerostin is an osteocyte-secreted glycoprotein that inhibits osteoblast formation by binding to LRP5/6 receptors and inhibiting the Wnt signalling pathway in osteoblasts [23,34]. Interestingly, western blotting and quantitative polymerase chain reaction (qPCR) analysis of bone lysate revealed a reduction in both protein and mRNA levels of sclerostin in *Tsc1* CKO mice compared with that in control mice (figure 4*a*,*b*). Immunostaining confirmed the reduction of sclerostin (figure 4*c*) in bone of *Tsc1* CKO mice, while immunofluorescence (IF) staining indicated an enhanced expression of β-catenin in osteoblasts (figure 4*d*). Hence, we consider the increased osteogenesis due to sclerostin-negative regulation of β-catenin expression in osteoblasts. Several studies have shown that sclerostin is expressed in MLO-Y4 osteocyte-like cells [35,36]. To further verify whether TSC1 controls sclerostin expression *in vitro*, MLO-Y4 osteocyte-like cells were transfected with TSC1 shRNA lentivirus. Consistent with the downregulation of TSC1, sclerostin protein and mRNA levels were decreased in MLO-Y4 cells transfected with TSC1 shRNA lentivirus (figure 4*e*,*f*). It has been shown that TSC1/2 regulates various cellular processes and protein expression through mTORC1-dependent or -independent mechanisms. To test whether the decreased sclerostin secretion was mTORC1 dependent, MLO-Y4 osteocyte-like cells were treated with rapamycin. Interestingly, after treatment with 1 nM rapamycin, both

protein and mRNA levels of SOST were increased (figure 4*g*,*h*). To further identify the role of mTORC1 in TSC-controlled SOST expression, we examined whether rapamycin was able to rescue the decrease in the SOST level in MLO-Y4 cells infected with TSC1 shRNA lentivirus. As expected, rapamycin enhanced the levels of SOST mRNA and protein in MLO-Y4 cells with TSC1 knockdown (figure 4*i*,*j*). These data suggest that TSC1 may inhibit SOST expression in osteocytes via an mTORC1-dependent mechanism.

## 2.5. Osteocyte TSC1 prevents osteogenesis through an increase of sclerostin secretion

Sclerostin functions as a strong inhibitor of bone formation by antagonizing Wnt signalling in osteoblasts [37]. Our study showed that SOST reduction activated the activity of β-catenin in *Tsc1* CKO mice. To investigate the potential role of sclerostin in osteocyte TSC1 deficiency-enhanced bone formation, we cultured MC3T3-E1 cells with conditional medium (CM) collected from MLO-Y4 osteocyte-like cells infected with TSC1 shRNA lentivirus (figure 5*a*). As expected, osteogenesis was significantly improved by CM from MLO-Y4 cells infected with TSC1 shRNA (ΔTSC1), as manifested by increased activity of alkaline phosphatase (ALP) (figure 5*b*), increased capacity to form mineralized nodules (figure 5*c*) and enhanced expression of OCN in the treated cells (figure 5*d*). Moreover, when MC3T3-E1 cells were incubated with CM from MLO-Y4 cells treated with rapamycin (ΔR), ALP activity (figure 5*e*), mineralized nodule formation (figure 5*f*) and OCN expression (figure 5*g*) were reduced. These findings suggested that the osteogenesis is negatively regulated by TSC1 in osteocytes through secretion of paracrine factors. To further define the role of sclerostin in this process, we supplemented the CM from MLO-Y4 cells treated with TSC1 shRNA with recombinant sclerostin (rhSCL, 50 ng ml$^{-1}$) or supplemented the medium from MLO-Y4 cells treated with rapamycin with anti-sclerostin antibody (scl-Ab, 50 ng ml$^{-1}$). We found that anti-sclerostin antibody rescued MLO-Y4 ΔRap medium-inhibited MC3T3-E1 osteogenesis (figure 5*h*–*j*), while recombinant sclerostin blocked MLO-Y4 ΔTSC1 medium-stimulated MC3T3-E1 osteogenesis (figure 5*k*–*m*). Based on these findings, we propose that osteocyte TSC1 likely prevents osteogenesis through increasing sclerostin secretion.

## 2.6. Mechanical loading activates mTORC1 to prevent sclerostin expression in osteocytes

It is well established that osteocytes sense mechanical stress and mediate the actions of osteoblasts to regulate bone homeostasis. Cyclic stretch functioning in osteocytes could enhance osteogenesis as well as bone nodule formation, and dominate in regulating cytoskeletal remodelling [38,39]. To investigate the effect of mechanical loading on mTORC1 signalling in osteocytes, we exposed MLO-Y4 to a 10% elongation cyclic stretching at 1 Hz for 48 h using a Flexcell Strain Unit. No damaged, dead or apoptotic cells were observed microscopically after generating a stretching loading on cultured MLO-Y4 cells. The orientation of cells in stretch-loaded culture plates was uniform, in contrast to the random distribution observed in non-loaded culture plates. The cell morphology of stretch-loaded cells was flatter than

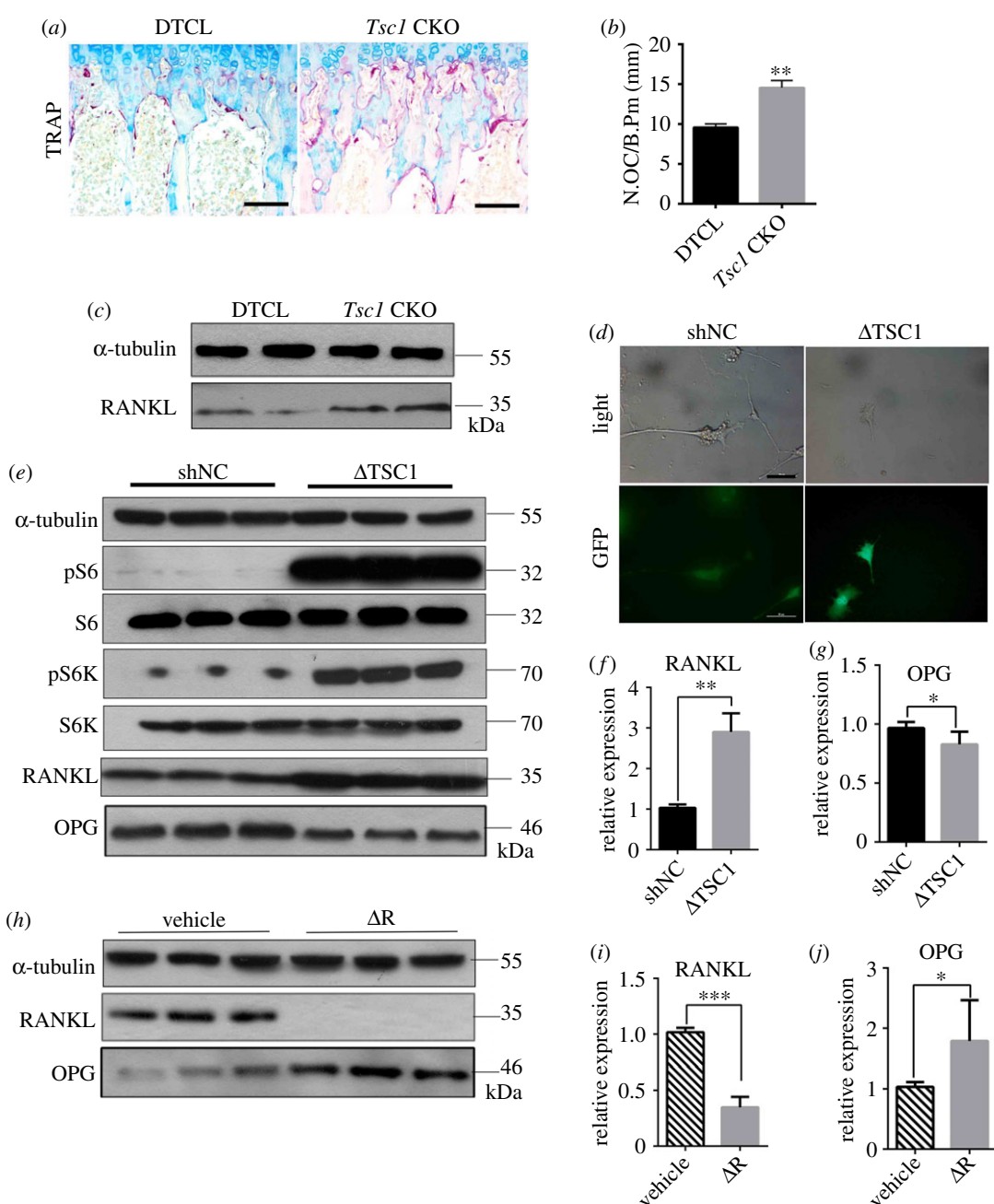

**Figure 3.** Activation of mTORC1 in osteocytes stimulates RANKL expression and osteoclast formation in mice. (*a*) Representative images of TRAP staining of femora from 10-week-old control (DTCL) and *Tsc1* CKO mice. The scale bars represent 50 μm. (*b*) Number of TRAP-positive osteoclasts (*t*-test, $p = 0.0017$, $n = 6$). (*c*) Western blotting analysis of RANKL in bone lysates from DTCL and *Tsc1* CKO mice ($n = 6$). (*d–g*) MLO-Y4 cells were infected with control shRNA lentivirus (shNC) and TSC1 shRNA lentivirus (ΔTSC1) for 72 h. (*d*) Representative images of lentivirus infection of MLO-Y4 cells. Scale bars, 50 μm. Western blot analysis of pS6, pS6K, RANKL and OPG (*e*) in MLO-Y4 cells infected with TSC1 shRNA lentivirus. qPCR analysis for (*f*) RANKL and (*g*) OPG mRNA levels (*t*-test, RANKL: $p = 0.0054$, OPG: $p = 0.0171$, $n = 6$). (*h–j*) MLO-Y4 cells were treated with 1 nM rapamycin (ΔR) and DMSO (vehicle) for 48 h, then cell lysates were subjected to immunoblotting (*h*) and qPCR analysis for (*i*) RANKL and (*j*) OPG protein and mRNA levels, respectively (*t*-test, RANKL: $p = 0.0003$, OPG: $p = 0.0220$, $n = 6$). All experiments were repeated independently three times. Data are represented as mean ± s.d., \*$p < 0.05$, \*\*$p < 0.01$ and \*\*\*$p < 0.001$.

that of cells grown in non-loaded plates (figure 6*a*), and the protein and mRNA expression levels of sclerostin were decreased in stretch-loaded MLO-Y4 cells compared with the non-loaded negative control (NC) group (figure 6*b,c*). These observations indicate that cyclic stretching suppresses sclerostin expression in osteocytes *in vitro*. Interestingly, the phosphorylation levels of both S6 and S6K1 were markedly increased in stretch-loaded cells (figure 6*b*), suggesting that mTORC1 activity is stimulated by mechanical loading in osteocytes *in vitro*. Furthermore, rapamycin could slightly affect stretch-loading inhibited sclerostin expression (figure 6*d,e*). These data implicate mechanical loading in the

suppression of sclerostin expression in osteocytes *in vitro* through activation of mTORC1.

## 2.7. TSC1 promotes sclerostin expression in osteocytes partially through Sirt1

Sirtuin 1 (Sirt1) is a histone deacetylase that acts as a novel bone regulator and represses sclerostin expression in osteocytes [40,41]. To explore the role of Sirt1 in TSC1/mTORC1-regulated sclerostin expression in osteocytes, we examined the level of Sirt1 in bone lysates from control and

royalsocietypublishing.org/journal/rsob Open Biol. 9: 180262

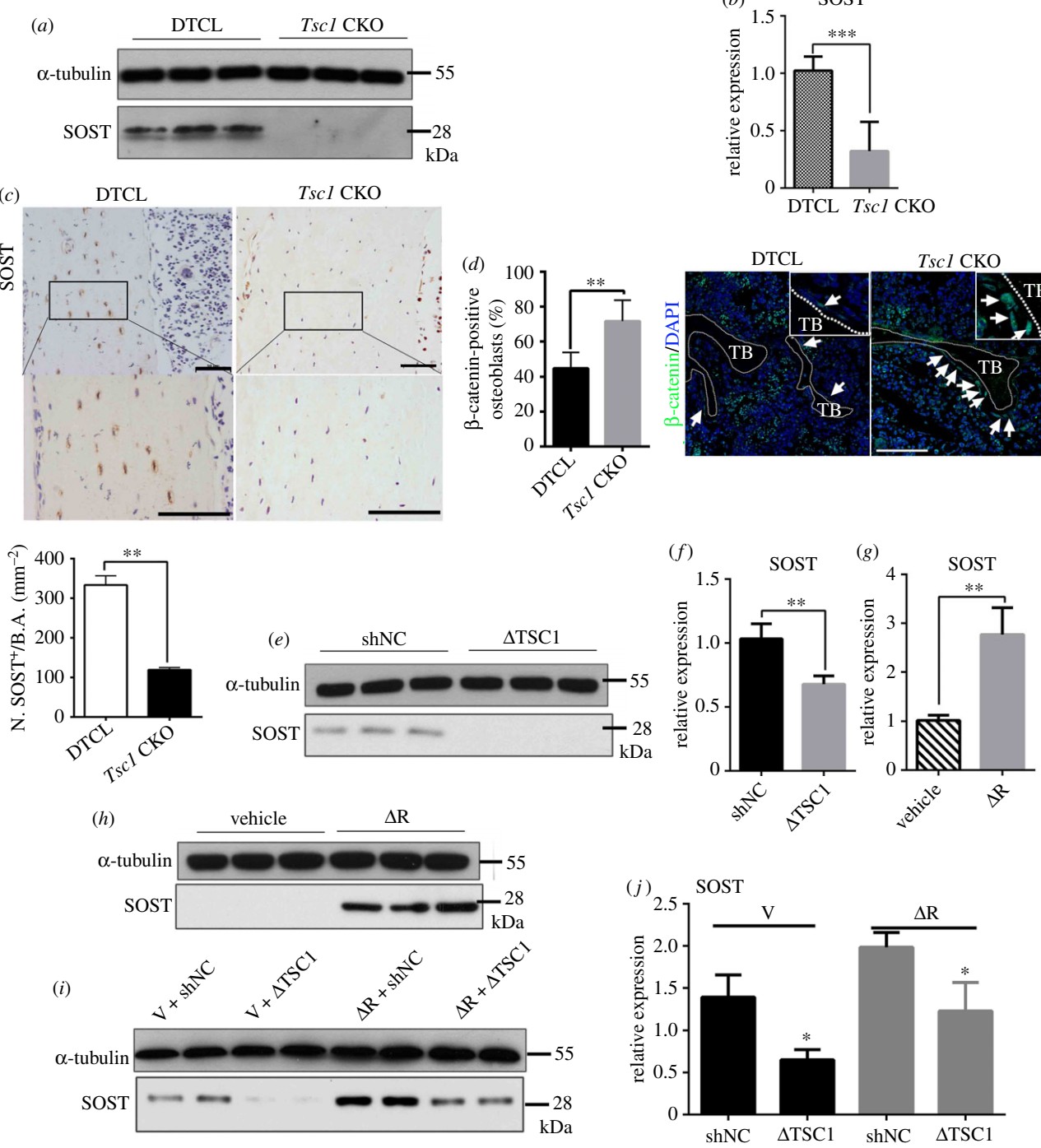

**Figure 4.** TSC1 inhibits sclerostin expression in osteocytes in an mTORC1-dependent manner. (*a,b*) The expression of (*a*) SOST protein and (*b*) mRNA (*t*-test, $p = 0.0005$) in bone lysates from 10-week-old control (DTCL) and *Tsc1* CKO mice ($n = 6$). (*c*) Immunohistochemical staining and quantification analysis of SOST in decalcified sections from the distal femora of 10-week-old male control (DTCL) and *Tsc1* CKO mice (*t*-test, $p = 0.0014$). The boxed area is magnified in the panel below. The scale bars represent 100 μm and 50 μm. SOST-positive cell numbers (N. SOST$^+$) between *Tsc1* CKO and control mice were analysed by cell counting. (*d*) Quantification analysis and immunofluorescence staining of β-catenin in decalcified sections from the distal femora of 10-week-old male control (DTCL) and *Tsc1* CKO mice (*t*-test, $p = 0.0078$). The image is magnified on the top right corner. TB, trabecular bone. The scale bar represents 50 μm ($n = 6$). (*e,f*) MLO-Y4 cells were infected with TSC1 shRNA lentivirus (ΔTSC1) for 72 h and then subjected to (*e*) immunoblotting and (*f*) qPCR analysis for SOST protein and mRNA levels, respectively (*t*-test, $p = 0.0052$). (*g,h*) MLO-Y4 cells were treated with vehicle and 1 nM rapamycin (ΔR) for 48 h and then subjected to (*g*) immunoblotting and (*h*) q-PCR analysis for SOST protein and mRNA levels, respectively (*t*-test, $p = 0.0029$). (*i,j*) MLO-Y4 cells were infected with NC and TSC1 shRNA lentivirus for 72 h, then treated with vehicle or 1 nM rapamycin for 48 h and subjected to (*i*) immunoblotting and (*j*) q-PCR analysis for SOST protein and mRNA levels, respectively (non-parametric statistical test, V group, $p = 0.0281$; ΔR group, $p = 0.0496$, $n = 6$). All experiments were repeated independently three times. Data are represented as mean ± s.d., *$p < 0.05$, **$p < 0.01$ and ***$p < 0.001$.

*Tsc1* CKO mice. We found that both protein and mRNA levels of Sirt1 were enhanced in *Tsc1* CKO mice (figure 7*a,b*). In cultured MLO-Y4 cells, infection of TSC1 shRNA lentivirus significantly induced Sirt1 expression (figure 7*c,d*) and accelerated *Sost* promoter H3K9 acetylation

(electronic supplementary material, figure S4A), while rapamycin treatment reduced Sirt1 levels (figure 7*e,f*). We next examined whether Sirt1 mediated the regulation of sclerostin by mTORC1. We found that the Sirt1 siRNA significantly increased the level of sclerostin (figure 7*g–i*) and slightly

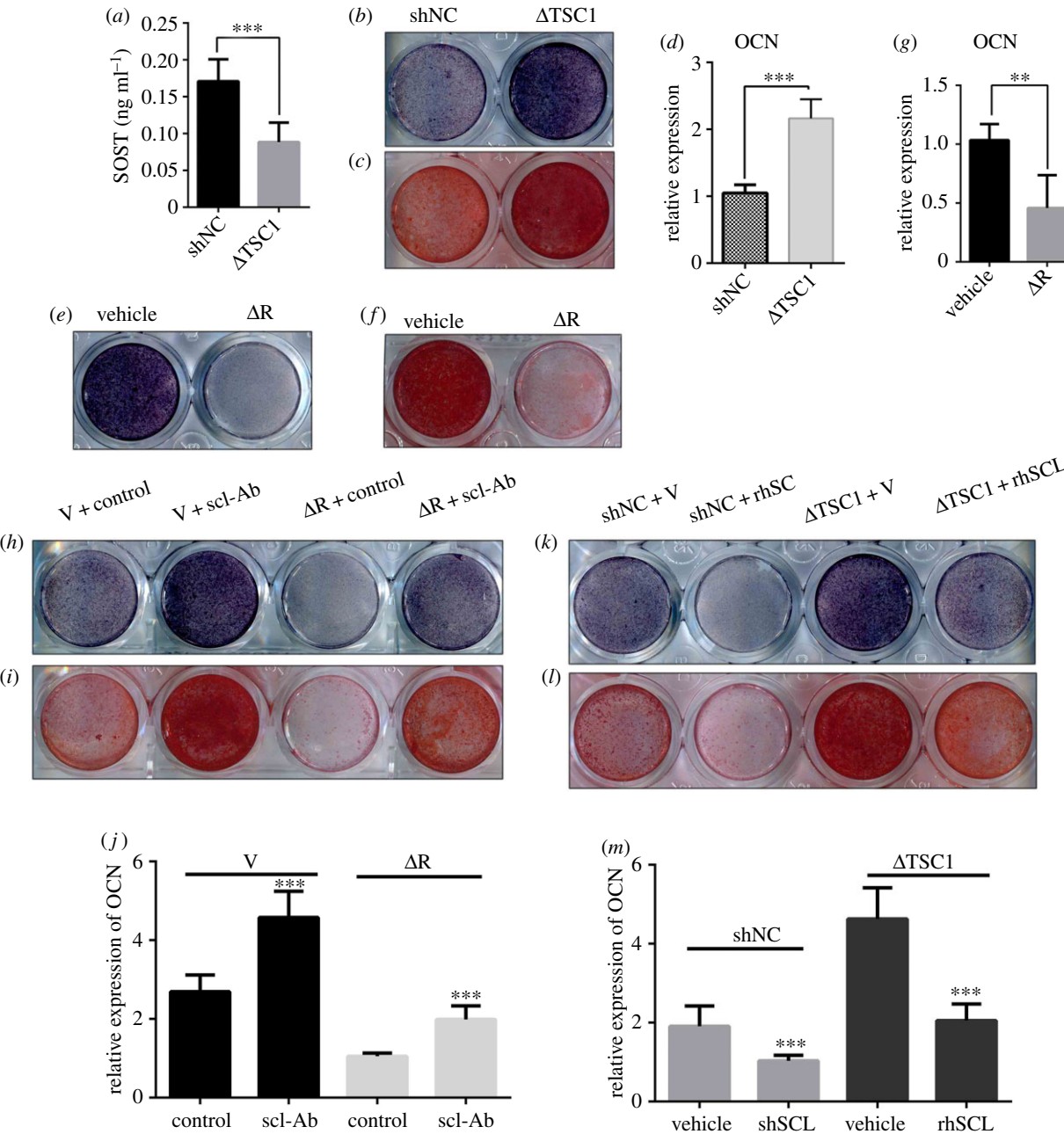

**Figure 5.** Osteocyte TSC1 prevents osteogenesis through increasing sclerostin secretion. (*a*) ELISA analysis of SOST in conditioned medium (CM) from MLO-Y4 cells infected with TSC1 or NC shRNA lentivirus for 72 h (*t*-test, $p = 0.0001$, $n = 6$). (*b–d*) MC3T3-E1 cells were incubated with CM from MLO-Y4 cells infected with TSC1 shRNA lentivirus ($\Delta$TSC1) and subjected to osteogenesis, and (*b*) ALP staining, (*c*) Alizarin red staining and (*d*) qPCR analysis of OCN (*t*-test, $p = 0.0009$, $n = 6$). (*e–g*) MC3T3-E1 cells were incubated with CM from MLO-Y4 cells treated with vehicle (V) and 1 nM of rapamycin ($\Delta$R) and subjected to osteogenesis and (*e*) ALP staining, (*f*) Alizarin red staining and (*g*) qPCR analysis of OCN (*t*-test, $p = 0.0038$, $n = 6$). (*h–j*) MC3T3-E1 cells were incubated with CM from MLO-Y4 cells treated with 1 nM rapamycin and anti-sclerostin antibody (scl-Ab, 50 ng ml$^{-1}$), as indicated, and subjected to osteogenesis and (*h*) ALP staining, (*i*) Alizarin red staining and (*j*) qPCR analysis of OCN (non-parametric statistical test, V group, $p = 0.0003$; $\Delta$R group, $p = 0.0005$, $n = 6$). (*k–m*) MC3T3-E1 cells were incubated with CM from MLO-Y4 cells infected with shNC and $\Delta$TSC1, with or without addition of recombinant sclerostin protein (rhSCL), as indicated (50 ng ml$^{-1}$), and subjected to osteogenesis and (*k*) ALP staining, (*l*) Alizarin red staining and (*m*) qPCR analysis of OCN. Data were analysed by a non-parametric statistical test; shNC group, $p = 0.0008$; $\Delta$TSC1 group, $p = 0.0005$, $n = 6$. All experiments were repeated independently three times. Data are represented as mean $\pm$ s.d., **$p < 0.01$ and ***$p < 0.001$.

affected sclerostin expression blocked by TSC1 shRNA lentivirus (figure 7*j,k*). Altogether, these findings suggest that activation of mTORC1 may prevent sclerostin expression in osteocytes, at least partially through Sirt1.

# 3. Discussion

The current study has established that TSC1 is important for sclerostin secretion in osteocytes and for osteogenesis in mice.

We demonstrate that ablation of TSC1 and activation of mTORC1 in osteocytes significantly reduces sclerostin production and stimulates osteogenesis both *in vitro* and *in vivo*. Our study also provides evidences to support the hypothesis that osteocyte TSC1/mTORC1 controls sclerostin secretion in response to mechanical loading, exerting its effects partially by modulating the expression of Sirt1 (figure 8).

Tuberous sclerosis, a mutation in TSC1/2, produces an autosomal dominant genetic disorder, characterized by

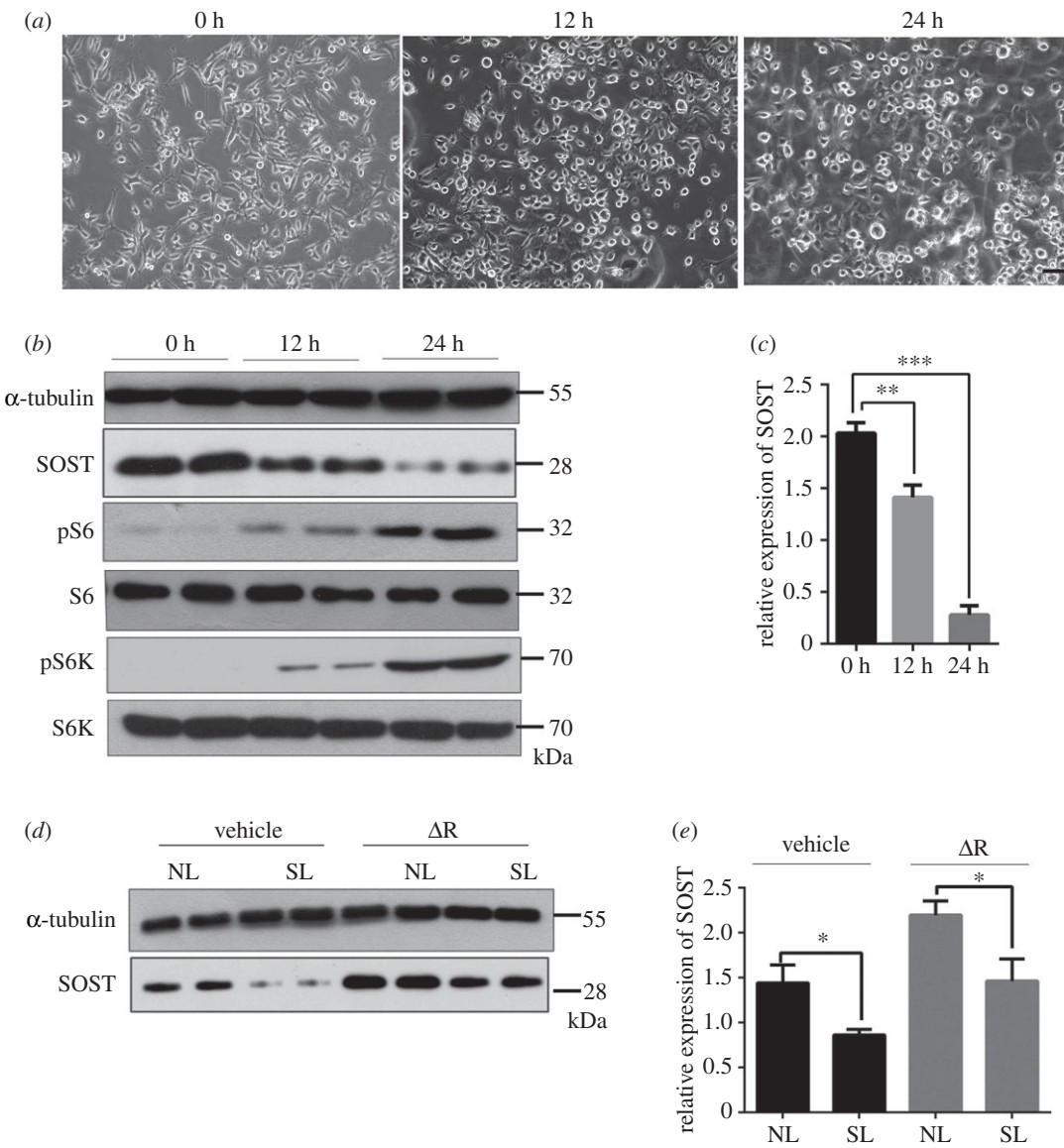

**Figure 6.** Mechanical loading activates mTORC1 to prevent sclerostin expression in osteocytes. (*a*) Representative images of the stretch-loaded MLO-Y4 cells for 0, 12 and 24 h. The scale bar represents 100 μm. 0 h: no-loaded group/control group; 12 h: stretch-loaded for 12 h; 24 h: stretch-loaded for 24 h. (*b*) Western blot analysis of SOST, pS6 and pS6K expression in MLO-Y4 cells stretch-loaded for 0, 12 or 24 h. (*c*) mRNA levels of SOST in stretch-loaded MLO-Y4 cells (one-way ANOVA with Bonferroni multiple comparison, 0 h versus 12 h, $p = 0.0077$; 0 h versus 24 h, $p = 0.0003$, $n = 6$). (*d,e*) MLO-Y4 cells were stretch-loaded and treated with 1 nM rapamycin for 24 h, then cell lysates were subjected to (*d*) western blotting or (*e*) qPCR analysis for SOST (non-parametric statistical test, vehicle group, $p = 0.0246$; ΔR group, $p = 0.0473$, $n = 6$). All experiments were repeated independently three times. NL, no-loaded; SL, stretch-loaded. Data are represented as mean ± s.d., *$p < 0.05$, **$p < 0.01$ and ***$p < 0.001$.

benign hamartomas in multiple organ systems [42]. The skeletal system is frequently involved in patients with TSC, who have sclerotic bone lesions such as increased skull density and thickened calvariae, but bone changes in a patient with TSC are not well understood [43]. Disruption of TSC2 in mature osteoblasts (*Oc*-Cre) or of TSC1 in preosteoblasts (*Osx*-Cre) resulted in high bone mass but damaged differentiation of osteoblasts and mineralized nodule formation in mice [29]. Intriguingly, ablation of TSC1 in osteocytes in this research also generated a phenotype with high bone mass [29]. TSC1 deletion activates mTORC1, promotes Sirt1 expression and downregulates sclerostin in osteocytes to accelerate bone formation in mice. Although these mechanisms remain to be clarified in patients with TSC, these mouse models may have potential implications for the cellular and molecular basis for the formation of sclerotic bone lesions in tuberous sclerosis.

Several studies have verified that mTORC1 is vital for cellular growth and development processes, and survival of osteoclasts and their progenitors, especially in macrophage colony-stimulating factor (M-CSF)- and RANKL-dependent mTORC1/S6K1 activity, as rapamycin reduced their survival *in vitro* [44]. Interestingly, we have found that deletion of TSC1 in murine B cells produces constitutive activation of mTORC1 and stimulates RANKL but represses OPG expression via negative regulation of AKT and β-catenin and subsequently promotes osteoclast formation and generates osteoporosis in mice [45]. Previous studies have confirmed that mTORC1/S6K1 could inhibit Akt via phosphorylation and degradation of IRS-1 and phosphorylation of Rictor (Thr1135) and inhibition of mTORC2 [46]. Although it has recently been demonstrated that osteocytes are a major source of RANKL to stimulate osteoclastogenesis during bone remodelling [47,48], how RANKL is regulated in

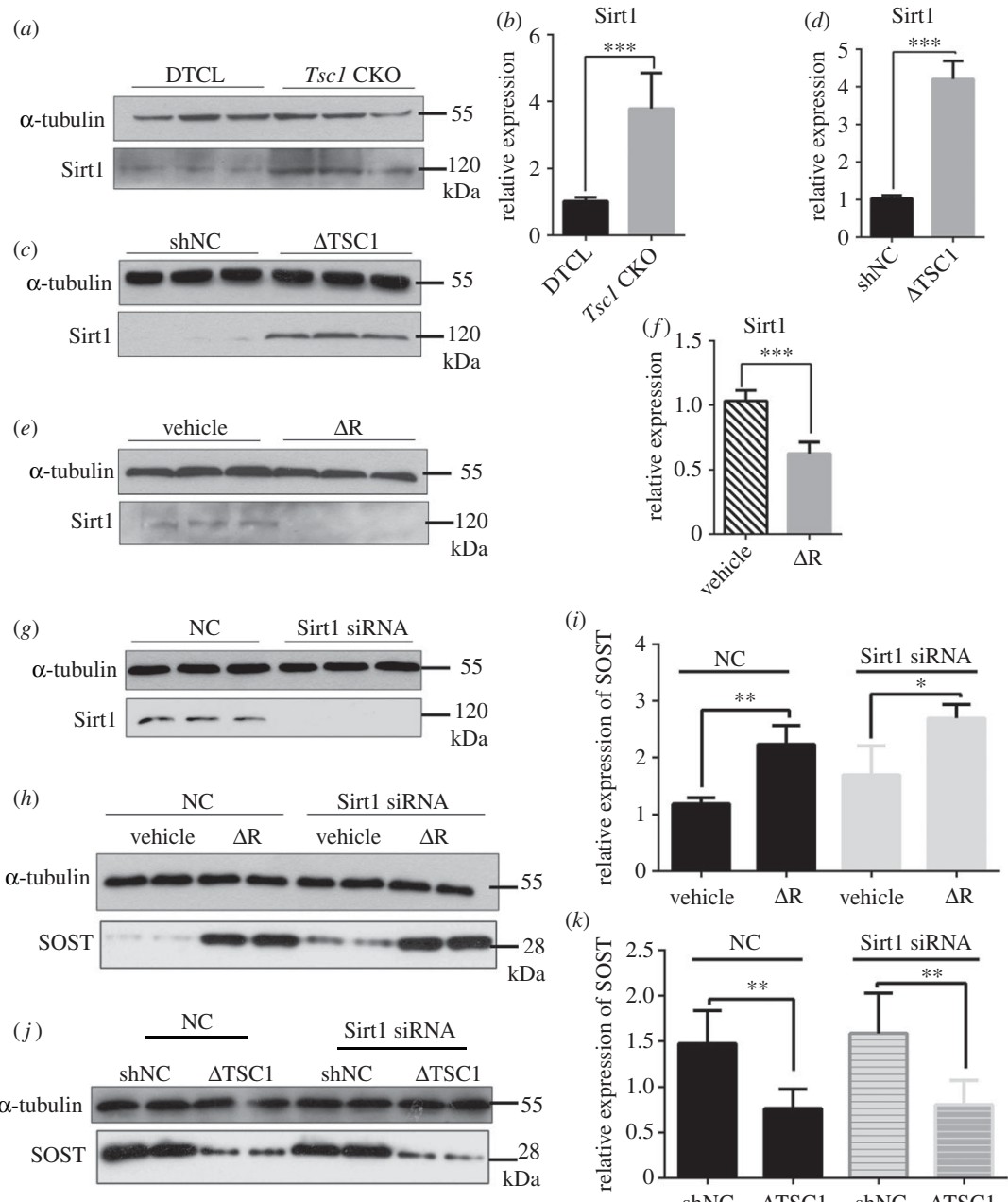

**Figure 7.** TSC1 promotes sclerostin expression in osteocytes partially through Sirt1. (*a*) Bone lysates from 10-week-old control and *Tsc1* CKO mice were subjected to western blot analysis for Sirt1. (*b*) mRNA expression of Sirt1 in femora of 10-week-old control (DTCL) and *Tsc1* CKO mice (*t*-test, $p = 0.0005$, $n = 6$). (*c*) Western blot analysis of Sirt1 expression in MLO-Y4 cells infected with TSC1 shRNA lentivirus. (*d*) mRNA levels of Sirt1 in MLO-Y4 cells infected with NC and TSC1 shRNA lentivirus (*t*-test, $p = 0.0009$, $n = 6$). (*e*) Western blot analysis of Sirt1 in MLO-Y4 cells treated with vehicle (V) and 1 nM of rapamycin for 48 h ($\Delta$R). (*f*) mRNA levels of Sirt1 in MLO-Y4 cells treated with 1 nM rapamycin for 48 h ($\Delta$R) (*t*-test, $p = 0.0008$, $n = 6$). (*g*) MLO-Y4 cells were transfected with Sirt1 or NC siRNA, (*h*) then treated with 1 nM rapamycin ($\Delta$R) for 48 h, and cell lysates were subjected to western blot analysis for SOST. (*i*) mRNA level of SOST was assayed by qPCR (non-parametric statistical test, NC group, $p = 0.0027$; Sirt1 siRNA group, $p = 0.0402$). (*j*) MLO-Y4 cells infected with TSC1 shRNA lentivirus ($\Delta$TSC1) for 72 h, subsequently transfected with Sirt1 or NC siRNA for another 48 h, then SOST expression was detected by western blotting. (*k*) mRNA level of SOST was assayed by qPCR (non-parametric statistical test, NC group, $p = 0.0033$; Sirt1 siRNA group, $p = 0.0021$). All experiments were repeated independently three times. Data are represented as mean $\pm$ s.d., *$p < 0.05$, **$p < 0.01$ and ***$p < 0.001$.

osteocytes remains unclear. Our results indicate that mTORC1 activation promotes RANKL expression in osteocytes and enhances osteoclast formation in bone, suggesting the regulation of RANKL by TSC1/mTORC1 in osteocytes. In addition to its anti-anabolic action, sclerostin is also capable of stimulating osteoclast differentiation in a RANKL-dependent manner and therefore has an indirect action in bone resorption [49]. Although sclerostin was down-regulated in *Tsc1* CKO mice, osteoclast formation in these mice was still enhanced. Our finding further supports the

theory that osteocyte-produced RANKL is essential for osteoclast formation in bone.

Antibodies against sclerostin protein reduce endogenous levels of sclerostin and restore BMD. This effect of the sclerostin antibody on bone has rendered it an important molecule in fracture healing, osteoporosis, metastatic disease and a variety of other disorders [50–52]. Recent evidence has shown that *Sost* gene expression is negatively regulated by Sirt1 through deacetylation of H3K9 in the promoter region [53]. Our study further indicates that TSC1/mTORC1

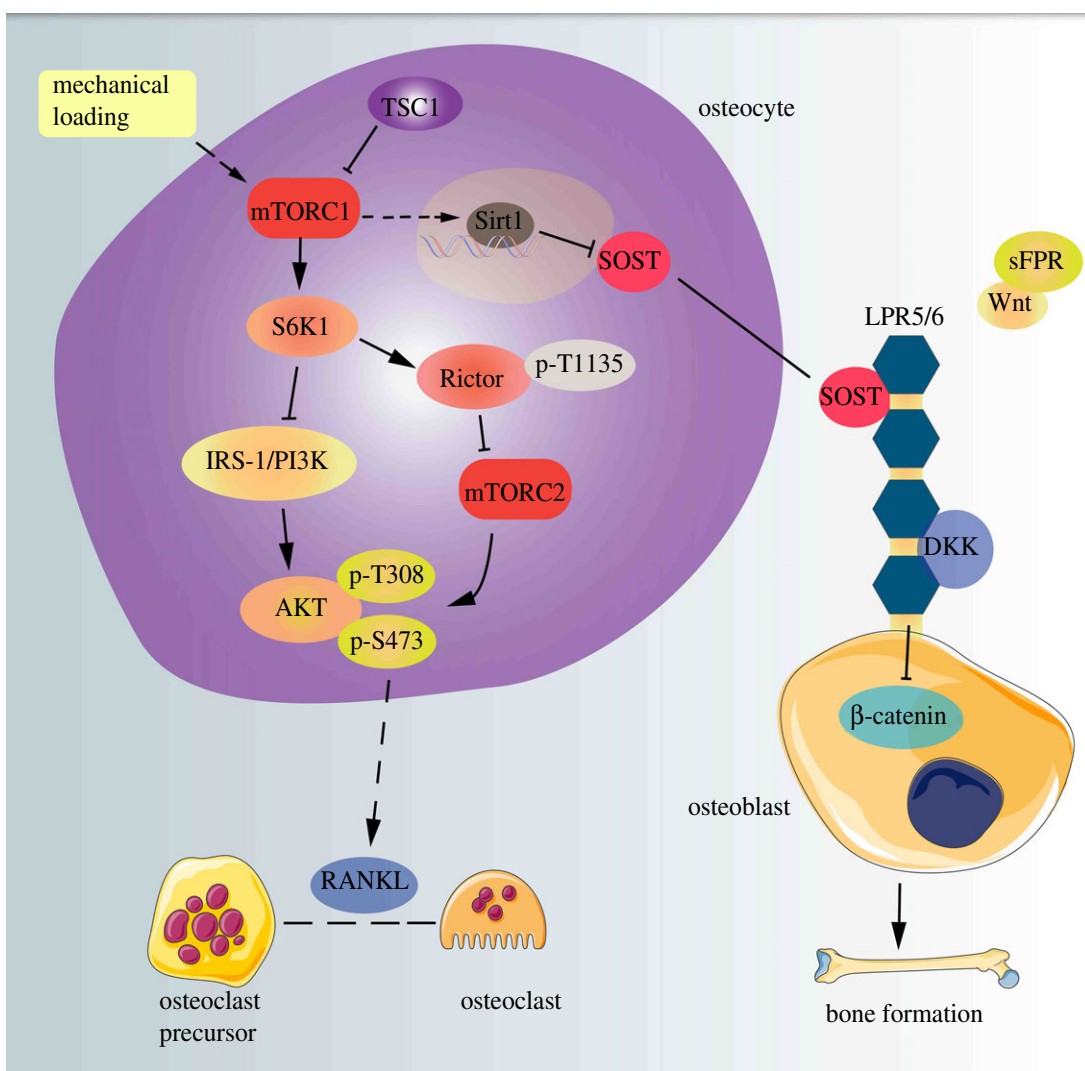

**Figure 8.** TSC1 prevents osteogenesis partially through promoting sclerostin secretion in osteocytes. A schematic model depicting the role of osteocyte TSC1 in the regulation of SOST and coordination of osteoclasts and osteoblasts.

regulates Sirt1 expression in osteocytes in response to mechanical loading. Targeting this pathway may present a strategy to increase osteogenesis and prevent bone loss-related diseases.

Although it has been established that osteocytes are key players in the response of bone to mechanical stimuli, it is still unclear whether their response to unloading is a direct response to a reduction in load as theorized by Wolff's law or a consequence of changes in systemic endocrine or paracrine factors [54]. Emerging evidence demonstrates that sclerostin expression is influenced by mechanical force stimulation, and mechanical loading is a key regulator of SOST transcription in the adult skeleton [55]. Unloading results in an increased number of SOST-positive osteocytes, and G-protein-coupled hormonal (PTH) and cytokine (PGE2) regulators are capable of suppressing the increases in sclerostin seen in mechanical unloading [56]. Here, we show that activation of mTORC1 and upregulation of Sirt1 may contribute to mechanical loading downregulation of sclerostin in osteocytes. Future therapies, aimed at modulating the gravity-sensing pathways of the osteocyte, could lead to improved efficacy for a range of bone disorders [57].

In summary, using a genetic approach, we clarified the role of TSC1/mTORC1 in controlling sclerostin secretion and bone formation (figure 8). Our present data may help to clarify the role of TSC1/mTORC1 in bone formation,

and importantly suggest that clinical therapy with mTOR inhibitors to retard bone loss may have underlying disadvantages. Pharmaceutical coordination of the pathway and agents in osteocytes may be beneficial in bone formation.

# 4. Material and methods

## 4.1. Experimental animals

All animal experiments were carried out with the approval of the Institutional Animal Care and Ethics Committee of Southern Medical University, Guangzhou, People's Republic of China. Importation, transport, housing and breeding of mice were performed according to the guidelines for the ethical treatment of animals. The *Tsc1*$^{f/f}$ (129S6/SvEvTac) and *DMP1*-Cre (JAX stock 023047) mice strains were purchased from Jackson Laboratory (Bar Harbor, ME, USA). *DMP1-Cre* mice express Cre recombinase under the control of the murine dentine matrix protein 1 (*DMP1*) promoter/enhancer elements. *Tsc1*$^{f/f}$ mice were backcrossed with C57BL/6 background mice (purchased from Southern Medical University Laboratory Animal Center) for eight generations before use. To generate TSC1-deficient mice in DMP1-expressing cells (mainly osteocyte lineage, few osteoblasts and gastrointestinal mesenchymal cells which have off-target effects [58]),

royalsocietypublishing.org/journal/rsob    Open Biol. 9: 180262

we mated $Tsc1^{f/f}$ mice with $DMP1$-Cre$^+$ mice to obtain $DMP1^+Tsc1^{f/+}$ mice, and then $DMP1^+Tsc1^{f/+}$ mice were bred with $Tsc1^{f/f}$ mice to obtain $DMP1^+Tsc1^{f/f}$ mice (termed $Tsc1$ CKO) and littermate controls ($DMP1^-Tsc1^{f/f}$ termed DTCL). Sex does affect skeletal development throughout growth in mammals. Therefore, we used all male mice for *in vivo* data in case of the interference of oestrogen. Male experimental and control littermate mice were compared at 4, 8 and 12 weeks of age. Genotyping was carried out on tail DNA by PCR using specific primers as follows: loxp$Tsc1$ allele forward, 5′-GTC ACG ACC GTA GGA GAA GC-3′ and reverse, 5′-GAA TCA ACC CCA CAG AGC AT-3′; $DMP1$-Cre forward, 5′-CCCGCAGAACCT-GAAGATG-3′ and reverse, 5′-GACCCGGCAAAACAGGTA G-3′. Food and water were available *ad libitum*. Mice were sacrificed by cervical dislocation.

## 4.2. Cell preparation

The murine-origin MLO-Y4 osteocyte-like cells were purchased from Gennio (Guangzhou, China). MLO-Y4 osteocyte-like cells were cultured in Dulbecco's modified Eagle medium (DMEM) (Gibco, 27515005) supplemented with 10% fetal bovine serum (FBS; Gibco) and 1% penicillin and streptomycin. Rapamycin (Sigma-Aldrich) and TSC1 shRNA lentivirus were added according to the manufacturer's protocol.

The preosteoblast cell line MC3T3-E1, purchased from ATCC (Manassas, VA, USA), was maintained in α-MEM (Gibco, 81117048) supplemented with 10% FBS (Gibco) and 1% penicillin and streptomycin at 37°C with 5% $CO_2$. After reaching confluence in 24-well plates, the medium was replaced with the CMT/CMR (conditioned medium collected from MLO-Y4 osteocyte-like cells infected with TSC1 shRNA lentivirus or supplemented with rapamycin). Then, 50 ng ml$^{-1}$ mouse sclerostin antibody (scl-Ab; Sigma, SAB1300086) or 50 ng ml$^{-1}$ recombinant human sclerostin (rhSCL; Novoprotein, Summit, NJ, USA; CD48) was added to the CMT/CMD as indicated. For osteogenic induction, 0.1 μM dexamethasone, 100 μg ml$^{-1}$ ascorbic acid and 10 mM β-glycerol phosphate (Sigma-Aldrich) were added to confluent cells. ALP staining and Alizarin red staining were carried out according to the manufacturer's instructions.

## 4.3. Mechanical loading

The MLO-Y4 osteocyte-like cells were divided into three groups: the control group (no loading to the cultured cells/ 0 h) and two treatment groups loaded over 12 or 24 h. These groups were seeded into six-well, flexible-bottomed plates at a density of $3 \times 10^5$ cells/well. After overnight incubation, when the cells had almost reached confluence, they were deprived of serum for 16 h before addition of CM. Treatment groups were cyclically strained with 10% strain at 1 Hz for 12 and 24 h in cell culture medium using the Flex-cell Strain Unit (Flexcell FX-5000T; Flexcell Corp., Burlington, NC, USA). The control group was cultured in similar plates and maintained in the same incubator without cyclic stretching. Three biological replicates were cultured and assayed for each analytical method.

## 4.4. Quantitative PCR and microarray analysis

The calvariae and femora as well as MLO-Y4 cell pellets were collected for qPCR analysis. Total RNA was collected with TRIZOL Reagent (Life Technologies; 15596–018) and extracted according to the manufacturer's instruction. Total RNA was transcribed into cDNA by using Prime Script Reverse Transcriptase (Takara; 2680B) at 100 ng in a total volume of 20 μl following the manufacturer's protocol, incubating at 37°C for 15 min, followed by heating at 85°C for 5 s. Two microlitres of cDNA was used as a template for qPCR using SYBR Premix Ex Taq (Takara; RR420A). qPCR reactions were performed using an ABI 7500 system (Applied Biosystems, Foster City, CA, USA). The specific sequences of primers used for this study are listed in electronic supplementary material, table S4, and the results are normalized to $GAPDH$ expression.

## 4.5. Western blotting analysis

The distal and proximal ends of the femur were removed and bone marrow cells were flushed out completely with phosphate-buffered saline (PBS). The surfaces of the bone shafts were scraped with a scalpel to remove the periosteum. Mouse tissues or cell cultures were lysed in RIPA buffer (50 mM Tris-HCl pH 7.4, 1% NP-40, 0.25% Na-deoxycholate, 150 mM NaCl, 1 mM EDTA, pH 7.4) with the addition of cocktails of protease inhibitors and phosphatase inhibitors (Roche, Basel, Switzerland), followed by centrifugation at 10 000$g$ for 20 min at 4°C. The supernatant was transferred to a new tube and the protein concentration was determined using a bicinchoninic acid assay. Protein samples were analysed by sodium dodecyl sulfate–polyacrylamide gel electrophoresis (SDS–PAGE) and transferred onto a nitrocellulose membrane (Bio-Rad Laboratories, Hercules, CA, USA). The membrane was then incubated with specific antibodies to α-tubulin (Santa Cruz Biotechnology, Santa Cruz, CA, USA; sc-8035, 1 : 2000), TSC1 (Cell Signaling Technology, 4906, 1 : 1000), phospho-S6K (T389) (Cell Signaling Technology, Danvers, MA, USA; 9234, 1 : 1000), S6K (Santa Cruz Biotechnology, sc-8418, 1 : 2000), phospho-S6 (S235/ 236) (Cell Signaling Technology, 2211, 1 : 1000), S6 (Santa Cruz Biotechnology, sc-74459, 1 : 2000), OCN (Santa Cruz Biotechnology, sc-23790, 1 : 2000), SOST (Sigma, SAB1300086, 1 : 500), RANKL (Boster, Wuhan, China; PB1064, 1 : 500), osteoprotegerin (Abcam, ab183910, 1 : 1000), Sirt1 (Proteintech, Rosemont, IL, USA; 13161-1-AP, 1 : 500) and α-tubulin as an internal control for protein loading. The membrane was then visualized by enhanced chemiluminescence using an ECL Kit (Amersham Biosciences, Chalfont St Giles, UK).

## 4.6. Enzyme-linked immunosorbent assay

Blood samples were collected from 12-week-old control and experimental mice after overnight fasting. The protein levels of P1NP and OCN released into the culture supernatants were measured using a mouse OCN enzyme-linked immunosorbent assay (ELISA) kit (MyBioSource, San Diego, CA, USA; MBS268470), mouse P1NP ELISA kit (Houston, TX, USA) and CTX-1 ELISA kit (CEA665Mu; Cloud-Clone Co., Ltd, Wuhan, China). The protein level of sclerostin from CM collected from MLO-Y4 osteocyte-like cells infected with TSC1 shRNA lentivirus was measured using a mouse

sclerostin ELISA kit (Boster, Wuhan, China), according to the manufacturer's instructions.

## 4.7. Histochemistry

Femurs dissected from DTCL and *Tsc1* CKO mice were embedded in paraffin after fixation in 4% paraformaldehyde in 0.1 M PBS (pH 6.5) at 4°C for 24 h and decalcification in 14% ethylenediaminetetraacetic acid (EDTA; pH 7.2) at 4°C for 14 days. After washing with running water for 2 h at room temperature, the decalcified femurs were dehydrated using graded ethanol, hyalinized by dimethylbenzene and then deposited in paraffin. The embedded femurs were sectioned into 3 μm sections along the longitudinal axis in preparation for histological analyses. After removal of the paraffin, H&E staining and immunohistochemistry were performed as previously described [59]. The sections were stained for H&E and TRAP (Sigma-Aldrich) activity and counter-stained with toluidine blue following a standard protocol on the product description.

## 4.8. Immunohistochemistry and immunofluorescence

Immunostaining was performed following a standard protocol. For immunohistochemistry, primary antibodies used in this study were diluted in the blocking buffer and the femur sections were incubated overnight at 4°C. The following specific primary antibodies and dilution were used: OCN (Abcam; ab93876, 1 : 200), DMP1 (Bioworld Technology, St Louis Park, MN, USA; BS7995, 1 : 200), pS6 (Ser235/236) (Cell Signaling Technology; 2211, 1 : 100), SOST (Sigma; SAB1300086, 1 : 200) and RANKL (Boster; PB1064, 1 : 200). After incubation of the primary antibodies, the femur sections were washed with PBS and sequentially incubated with horseradish peroxidase (HRP)-loaded secondary antibodies for 1 h at room temperature. The secondary antibodies were HRP goat anti-rabbit IgG antibodies (Abcam; ab6721, 1 : 200). All sections were observed and photographed on an Olympus BX51 microscope. Osteocytes in osseous tissue were evaluated by morphology and calculated by two independent observers blinded to the groups. In immunohistochemistry assays, cells per bone area (B. Ar) were used to calculate the number of positive cells, and integrated optical density per area of positive cells (IOD/area, mean density) was used to quantify the staining intensity by detecting cells in six different images taken at 100× magnification with Image Pro Plus 6.0 software (Media Cybernetics, MD, USA). In brief, positively stained regions of the image were selected by HIS (hue, saturation and lightness) with H in the range 0–25, I in the range 0–210 and S in the range 0–255, and then the brown colour was converted into a greyscale signal. The greyscale signal was measured as the mean optical intensity of staining (mean density) within the tissue masks. At least three mice per group were examined. Three equidistant sections spaced at 200 μm apart throughout the midsagittal section of femur were evaluated. For IF, we incubated primary antibodies against β-catenin (1 : 100) overnight at 4°C, and then we washed the femur sections with PBS. Sequentially, secondary fluorescent antibodies (1 : 500; Invitrogen) were added and slides were incubated at room temperature for 60 min in the dark. After incubation with secondary antibodies, slides were washed with PBS, stained with DAPI in the dark and coverslipped. IF images were collected using a confocal laser scanning microscope (Olympus, Tokyo, Japan). β-catenin was labelled green and nuclei were stained blue.

## 4.9. Histology of non-decalcified bone

Undecalcified histomorphometric analysis was performed on femurs of 10-week-old mice through intraperitoneal injection with calcein (Sigma; 15 mg kg$^{-1}$ body weight) in 2% sodium bicarbonate solution 10 and 3 days before sacrifice. The dissected femurs were embedded in methylmethacrylate after fixation in 4% paraformaldehyde for 24 h and dehydration through a graded series of ethanol (70–100%) and xylene, without decalcification. Sections 10 μm thick were used for double-labelling fluorescent analysis. Representative images were captured using an LSM510 Meta confocal microscope (Zeiss, Oberkochen, Germany). For dynamic histomorphometry, MAR, MS/BS and BFR of the femoral cortex were measured by the OsteoMeasure morphometry system (OsteoMetrics Inc., Decatur, GA, USA).

## 4.10. Bone parameter microcomputed tomography assay

The long bones or vertebrae were dissected free of soft tissue from 4-, 8-, 12-week-old mice, fixed in 70% ethanol and analysed using a Scanco micro-CT-80 scanner (Scanco Medical, Bassersdorf, Switzerland). The samples were scanned at a voltage of 75 kV and a resolution of 12 μm per pixel. The three-dimensional structure was constructed through a total of 100 sections at the primary trabecular bone of the metaphysis. For trabecular morphometric analysis, BMD, bone volume/tissue volume (BV/TV), trabecular number (Tb. N), trabecular thickness (Tb. Th) and trabecular separation (Tb. Sp) were performed using a native analysis system of the micro-CT to quantify a total of 100 sections at the primary trabecular bone of the lower femoral metaphysis as areas of interests. For analysis of femoral cortex evaluation, the cortical thickness (Ct. Th) and periosteal perimeter (Ps. Pm) were assessed using the same system as described above, with 100 sections of the mid-shaft femur. A total of 50 sections of the mid-shaft spinal column were chosen as areas of interests to calculate the trabecular statistics of the spinal column.

## 4.11. siRNA knockdown

We transiently transfected MLO-Y4 osteocyte-like cells with Sirt1 siRNA using Lipofectamine RNAi MAX (Invitrogen, Carlsbad, CA, USA) in Opti-MEM medium (Invitrogen), according to the manufacturer's instructions. The sequence of Sirt1 siRNA was as follows: Sirt1 F, 5′-GCACCGAUCCUC-GAACAAUTT-3′; Sirt1 R, 5′-AUUGUUCGAGGAUCGGUG CTT-3′. Universal NC siRNA was used as the control for non-sequence-specific effects. The efficiency of siRNA knockdown was confirmed by western blotting and qPCR. Each experiment was performed in triplicate.

## 4.12. shRNA lentivirus

The lentivirus for short hairpin (shRNA)-mediated knockdown of TSC1 (shTSC1; 5′-GACACACAGAATAGC-TATG-3′) and non-silence control lentivirus (shNC; 5′-TTCTCCGAACGTGTCACGT-3′) were purchased from Shanghai GenePharma Co. Ltd (Shanghai, China). For viral

infection, the optimum multiplicity of infection was obtained before shRNA lentivirus infection. In order to establish the stable cell line, MLO-Y4 osteocyte-like cells were plated into six-well plates and cultured until reaching approximately 70% confluence then infected with lentivirus in the presence of 5 $\mu$g ml$^{-1}$ polybrene (Sigma). GFP expression was detected to assess the infection efficiency 3 days after infection (figure 4$d$). Then the growth medium containing 3 $\mu$g ml$^{-1}$ of puromycin was used to select the infected cells for one week before using them for subsequent experiments. Expression efficiency was evaluated by western blot analysis.

## 4.13. Statistics

All quantitative data are presented as mean $\pm$ s.d. with a minimum of three independent samples. Statistical significance in each group was analysed using the unpaired, two-tailed Student's $t$-test or a one-way analysis of variance (ANOVA) with the Bonferroni *post hoc* test. The level of significance was set at $*p < 0.05$, $**p < 0.01$ and $***p < 0.001$.

**Ethics.** All animal experiments were carried out with the approval of the Institutional Animal Care and Ethics Committee of Southern Medical University. Mice importing, transporting, housing and breeding were performed according to the guidelines for the ethical treatment of animals.

**Data accessibility.** Supporting data can be found in the electronic supplementary material.

**Authors' contributions.** X.B., A.L. and Z.-K.C. conceived the study and analysed and interpreted the data. M.L., Z.Z., Ji.Y. and G.X. provided samples. W.L., Z.W. and Ju.Y. performed the histological review. W.L. and Z.W. performed the statistical analysis. W.L., Y.W., K.L., B.H., B.Y. and T.W. acquired the data. W.L. and Z.W. contributed equally to this article. All authors meet the International Committee of Medical Journal Editors recommendations and have critically reviewed the manuscript.

**Competing interests.** The authors declare that they have no conflicts of interest with the contents of this article.

**Funding.** This work was supported by grants from the National Natural Science Foundation of China (81871745, 31800840, 81625015 and 31529002), the State Key Development Program for Basic Research of China (2015CB553602), and the Program for Changjiang Scholars and Innovative Research Team in University (IRT_16R37).

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
