## [Reviewer comments · Open Biology]

Review History

RSOB-18-0262.R0 (Original submission)

Review form: Reviewer 1

Recommendation

Major revision is needed (please make suggestions in comments)

Are each of the following suitable for general readers?

- a) **Title**
Yes
- b) **Summary**
Yes
- c) **Introduction**
Yes

Is the length of the paper justified?

Yes

Should the paper be seen by a specialist statistical reviewer?

No

Is it clear how to make all supporting data available?

Yes

Is the supplementary material necessary; and if so is it adequate and clear?

Yes

Do you have any ethical concerns with this paper?

No

Comments to the Author

The paper outlines a set of well-designed studies aimed at providing evidence that TSC1 regulates sclerostin expression in osteocytes. Sclerostin is a potent antagonist of bone formation and has become a strong player as a treatment target to increase bone mass in patients with osteoporosis and other disease which compromise the skeleton. This is an interesting area in the field of bone biology as uncovering new targets which regulate sclerostin could provide new avenues to exploit this pathway to rebuild bones mass. The use of in vitro and in vivo studies provides strong evidence for this regulation, although the studies are well executed, the quality of some of the results presented is not however adequate for publication, as outlined below. Further, a number of errors are evident throughout the paper in regards to axis labels, figure labels and references to figure elements, as outlined below. Lastly, the authors should ensure they have adequately referenced the literature in this area as some relevant findings have not been discussed, as outlined below. They should also discuss the fact that their data pertaining to mechanical regulation of sclerostin expression through activation mTORC1 is only in vitro and requires in vivo validation. To be certain that this is a mechanism, TSC1 KO mice should be loaded to see if the response to load is suppressed. In vivo other mechanisms may compensate, these cannot be explored in vitro.

Major items:

- 1) Reference should be made to papers discussing HDAC5 (Li 2016) and PTH (Wein 2015) as upstream regulators of sclerostin expression by osteocytes in the introduction or discussion
- 2) In results section on uCT data the authors state that the mice show progressive osteoporosis. This term refers to reduced resorption and given the mechanism for the increased bone mass is not determined by this point in the paper it should be replaced with increased bone mass or volume. Also it states that trabecular separation is increased when it is actually decreased.
- 3) The BV/TV data provided in table S1-3 does not denote if it is from femur or vertebra and also the values for BV/TV seem oddly low in the WT mice. Are these values in %?
- 4) Figure 2F, MAR values are odd, they are unusually high so it is possible the data has not been divided by the days between the labels, 7 days.
- 5) Figure S3D the label on the graph does not match how the data is described
- 6) Osteoclast analysis is not conclusive enough. The number of TRAP+ cells should be expressed as a % of bone surfaces, is this the case? It is not clear from the results of methods. Further, both in vivo and in vitro analysis of osteoclasts did not examine their function. This should be examined using serum CTX from mice or examine resorptive capacity on calcium phosphate discs or dentine. Functional analysis is required to truly confirm the impact of TSC1 deletion on bone resorption. The increased osteoclast number may just be a feedback response given bone formation is up. This should be discussed. Further assessment of serum RANKL and OPG would

help determine if the changes in this pathway are only occurring locally to the osteocyte or is it widespread through changes in circulating levels?

7) A number of protein blots in the paper are not adequate for publication. These include fig 4H and I, 6B and 6D, 7 H and J. These western blots should be repeated and cleaner data presented.

8) Figure 6A is difficult to appreciate that the cell morphology changed with loading as there is no image of a non-loaded plate of cells.

9) Page 13 reference to rapamycin rescuing the stretch-loading inhibition of sclerostin is incorrect based on the data in figure 6B. Unless the reviewer is interpreting the figure incorrectly, SOST is still significantly reduced with rapamycin treatment. The fact that in all of the figure legends fail to confirm the comparison being made for each ** symbol does not help with this confusion.

10) Reference to figure 7J and K, Sirt1 siRNA did not appear to rescue scl expression blocked by TSC1 shRNA, but this is the interpretation by authors. Again the ** reference is not clear and may make this clearer to the reader.

11) All in vivo data presented is showing male mice. Did female mice show a similar phenotype? This should be discussed as sex can manifest its own impact on the skeleton.

Minor items:

Page 5 first paragraph, monoclonal antibodies to sclerostin inhibit its activity but do not reduce sclerostin. This should be clarified/ Also

Page 7, paragraph 1 last line "Altogether, these data suggest that TSC1 deletion and mTORC1 activation in osteocyte markedly increase bone mass in mice. Should read Altogether, these data suggest that TSC1 deletion and mTORC1 activation in osteocytes markedly increase bone mass in mice.

Page 9 line 3 showed a dramatic increase of RANKL....this change is not that obvious so dramatic feels like and over statement here.

Figures 5I and 5L are not referred to anywhere in the text of the paper, this should be corrected
Page 14 line 2, reference to figure 7G and I should be 7G and H.

Review form: Reviewer 2

Recommendation

Accept with minor revision (please list in comments)

Are each of the following suitable for general readers?

- a) **Title**
Yes
- b) **Summary**
Yes
- c) **Introduction**
Yes

Is the length of the paper justified?

Yes

Should the paper be seen by a specialist statistical reviewer?

Yes

Is it clear how to make all supporting data available?

Yes

Is the supplementary material necessary; and if so is it adequate and clear?

No

Do you have any ethical concerns with this paper?

No

Comments to the Author

The manuscript by Liu et al examines the role of conditional deletion of tuberous sclerosis complex 1 (TSC1) in the skeletal homeostasis. Liu et al show that in growing mice, conditional deletion of TSC1 results in a high bone remodeling phenotype with a positive balance that leads to bone gain. In addition, they found that mice lacking TSC1 in osteocytes have decreased expression of Sost/Sclerostin and increase expression of Rankl. Mechanistic studies showed that TSC1 activates mTORC1 signaling, which in turn activates Sirt to downregulate Sost/Scl expression and favor osteoblast differentiation. Further, authors found that mTORC1 signaling is also involved in the regulation of Sost by mechanical stimulation using in vitro approaches. This work presents a remarkable bone phenotype induced by deletion of TSC1 in osteocytes. The studies are technically very well done and the work is likely to be of significant interest to a wide range of researchers. However, here are some issues that need to be modified before the manuscript can be considered for publication.

1. "In summary, using a genetic approach, we clarified the role of TSC1/mTORC1 in controlling sclerostin secretion and bone formation (Figure 9)" in this part should be changed to Figure 8.
2. In Figure legend part, H should be changed to G in Figure 1.
3. In Figure 5 J, vertical ordinate expression should be changed to relative expression of OCN.
4. In Figure 5 M, vertical ordinate expression should be changed to relative expression of OCN.
5. The study claims that osteoclastogenesis and osteogenesis are both enhanced in Tsc1CKO mice. Please clarify the mechanism of osteoclast activities and its affect on bone formation.

Decision letter (RSOB-18-0262.R0)

25-Feb-2019

Dear Professor Bai,

We are writing to inform you that the Editor has reached a decision on your manuscript RSOB-18-0262 entitled "Osteocyte TSC1 promotes sclerostin secretion to restrain osteogenesis in mice", submitted to Open Biology.

As you will see from the reviewers' comments below, there are a number of criticisms that prevent us from accepting your manuscript at this stage. The reviewers suggest, however, that a revised version could be acceptable, if you are able to address their concerns. If you think that you can deal satisfactorily with the reviewer's suggestions, we would be pleased to consider a revised manuscript.

The revision will be re-reviewed, where possible, by the original referees. As such, please submit the revised version of your manuscript within six weeks. If you do not think you will be able to meet this date please let us know immediately.

When submitting your revised manuscript, please respond to the comments made by the referee(s) and upload a file "Response to Referees" in "Section 6 - File Upload". You can use this to document any changes you make to the original manuscript. In order to expedite the processing of the revised manuscript, please be as specific as possible in your response to the referee(s).

Please see our detailed instructions for revision requirements
<https://royalsociety.org/journals/authors/author-guidelines/>

Sincerely,

The Open Biology Team
mailto: openbiology@royalsociety.org

Reviewer(s)' Comments to Author(s):

Referee: 1

Comments to the Author(s)

The paper outlines a set of well-designed studies aimed at providing evidence that TSC1 regulates sclerostin expression in osteocytes. Sclerostin is a potent antagonist of bone formation and has become a strong player as a treatment target to increase bone mass in patients with osteoporosis and other disease which compromise the skeleton. This is an interesting area in the field of bone biology as uncovering new targets which regulate sclerostin could provide new avenues to exploit this pathway to rebuild bones mass. The use of in vitro and in vivo studies provides strong evidence for this regulation, although the studies are well executed, the quality of some of the results presented is not however adequate for publication, as outlined below. Further, a number of errors are evident throughout the paper in regards to axis labels, figure labels and references to figure elements, as outlined below. Lastly, the authors should ensure they have adequately referenced the literature in this area as some relevant findings have not been discussed, as outlined below. They should also discuss the fact that their data pertaining to mechanical regulation of sclerostin expression through activation mTORC1 is only in vitro and requires in vivo validation. To be certain that this is a mechanism, TSC1 KO mice should be loaded to see if the response to load is suppressed. In vivo other mechanisms may compensate, these cannot be explored in vitro.

Major items:

- 1) Reference should be made to papers discussing HDAC5 (Li 2016) and PTH (Wein 2015) as upstream regulators of sclerostin expression by osteocytes in the introduction or discussion
- 2) In results section on uCT data the authors state that the mice show progressive osteopetrosis. This term refers to reduced resorption and given the mechanism for the increased bone mass is not determined by this point in the paper it should be replaced with increased bone mass or volume. Also it states that trabecular separation is increased when it is actually decreased.
- 3) The BV/TV data provided in table S1-3 does not denote if it is from femur or vertebra and also the values for BV/TV seem oddly low in the WT mice. Are these values in %?
- 4) Figure 2F, MAR values are odd, they are unusually high so it is possible the data has not been divided by the days between the labels, 7 days.
- 5) Figure S3D the label on the graph does not match how the data is described
- 6) Osteoclast analysis is not conclusive enough. The number of TRAP+ cells should be expressed as a % of bone surfaces, is this the case? It is not clear from the results of methods. Further, both in vivo and in vitro analysis of osteoclasts did not examine their function. This should be examined using serum CTX from mice or examine resorptive capacity on calcium phosphate discs or dentine. Functional analysis is required to truly confirm the impact of TSC1 deletion on bone resorption. The increased osteoclast number may just be a feedback response given bone formation is up. This should be discussed. Further assessment of serum RANKL and OPG would help determine if the changes in this pathway are only occurring locally to the osteocyte or is it widespread through changes in circulating levels?
- 7) A number of protein blots in the paper are not adequate for publication. These include fig 4H and I, 6B and 6D, 7 H and J. These western blots should be repeated and cleaner data presented.
- 8) Figure 6A is difficult to appreciate that the cell morphology changed with loading as there is no image of a non-loaded plate of cells.
- 9) Page 13 reference to rapamycin rescuing the stretch-loading inhibition of sclerostin is incorrect based on the data in figure 6B. Unless the reviewer is interpreting the figure incorrectly, SOST is still significantly reduced with rapamycin treatment. The fact that in all of the figure legends fail to confirm the comparison being made for each ** symbol does not help with this confusion.
- 10) Reference to figure 7J and K, Sirt1 siRNA did not appear to rescue scl expression blocked by TSC1 shRNA, but this is the interpretation by authors. Again the ** reference is not clear and may make this clearer to the reader.
- 11) All in vivo data presented is showing male mice. Did female mice show a similar phenotype? This should be discussed as sex can manifest its own impact on the skeleton.

Minor items:

Page 5 first paragraph, monoclonal antibodies to sclerostin inhibit its activity but do not reduce sclerostin. This should be clarified/ Also

Page 7, paragraph 1 last line "Altogether, these data suggest that TSC1 deletion and mTORC1 activation in osteocyte markedly increase bone mass in mice. Should read Altogether, these data suggest that TSC1 deletion and mTORC1 activation in osteocytes markedly increase bone mass in mice.

Page 9 line 3 showed a dramatic increase of RANKL....this change is not that obvious so dramatic feels like and over statement here.

Figures 5I and 5L are not referred to anywhere in the text of the paper, this should be corrected

Page 14 line 2, reference to figure 7G and I should be 7G and H.

Referee: 2

Comments to the Author(s)

The manuscript by Liu et al examines the role of conditional deletion of tuberous sclerosis complex 1 (TSC1) in the skeletal homeostasis. Liu et al show that in growing mice, conditional deletion of TSC1 results in a high bone remodeling phenotype with a positive balance that leads to bone gain. In addition, they found that mice lacking TSC1 in osteocytes have decreased expression of Sost/Sclerostin and increase expression of Rankl. Mechanistic studies showed that TSC1 activates mTORC1 signaling, which in turn activates Sirt to downregulate Sost/Scl expression and favor osteoblast differentiation. Further, authors found that mTORC1 signaling is also involved in the regulation of Sost by mechanical stimulation using in vitro approaches. This work presents a remarkable bone phenotype induced by deletion of TSC1 in osteocytes. The studies are technically very well done and the work is likely to be of significant interest to a wide range of researchers. However, here are some issues that need to be modified before the manuscript can be considered for publication.

1. "In summary, using a genetic approach, we clarified the role of TSC1/mTORC1 in controlling sclerostin secretion and bone formation (Figure 9)" in this part should be changed to Figure 8.
2. In Figure legend part, H should be changed to G in Figure 1.
3. In Figure 5 J, vertical ordinate expression should be changed to relative expression of OCN.
4. In Figure 5 M, vertical ordinate expression should be changed to relative expression of OCN.
5. The study claims that osteoclastogenesis and osteogenesis are both enhanced in Tsc1CKO mice. Please clarify the mechanism of osteoclast activities and its affect on bone formation.

Author's Response to Decision Letter for (RSOB-18-0262.R0)

See Appendix A.

RSOB-18-0262.R1 (Revision)

Review form: Reviewer 1

Recommendation

Accept as is

Are each of the following suitable for general readers?

- a) **Title**
Yes
- b) **Summary**
Yes
- c) **Introduction**
Yes

Is the length of the paper justified?

Yes

Should the paper be seen by a specialist statistical reviewer?

No

Is it clear how to make all supporting data available?

Yes

Is the supplementary material necessary; and if so is it adequate and clear?

Yes

Do you have any ethical concerns with this paper?

No

Comments to the Author

The manuscript is markedly improved and the author have responded well to all requests. I feel it can be published in this format without further edits.

Decision letter (RSOB-18-0262.R1)

18-Apr-2019

Dear Professor Bai,

We are pleased to inform you that your manuscript entitled "Osteocyte TSC1 promotes sclerostin secretion to restrain osteogenesis in mice" has been accepted by the Editor for publication in Open Biology.

If applicable, please find the referee comments below. No further changes are recommended.

Sincerely,

The Open Biology Team
mailto:openbiology@royalsociety.org

Reviewer(s)' Comments to Author:

Referee:

Comments to the Author(s)

The manuscript is markedly improved and the author have responded well to all requests. I feel it can be published in this format without further edits.

Appendix A

Dear Prof. Glover,

On behalf of all co-authors, I would like to thank you for offering us the opportunity to resubmit our manuscript. We appreciate the editor's and reviewers' positive and constructive comments and suggestions on our manuscript entitled "Osteocyte TSC1 promotes sclerostin secretion to restrain osteogenesis in mice".

We have carefully considered the reviewers' comments and made revisions which are highlighted in red in our revised manuscript. Please find the revised version attached, which we would like to submit for your kind consideration. The revisions made to the paper and the responses to the editor's and reviewers' comments are as follows:

Referee: 1

Comments to the Author(s)

The paper outlines a set of well-designed studies aimed at providing evidence that TSC1 regulates sclerostin expression in osteocytes. Sclerostin is a potent antagonist of bone formation and has become a strong player as a treatment target to increase bone mass in patients with osteoporosis and other disease which compromise the skeleton. This is an interesting area in the field of bone biology as uncovering new targets which regulate sclerostin could provide new avenues to exploit this pathway to rebuild bones mass. The use of in vitro and in vivo studies provides strong evidence for this regulation, although the studies are well executed, the quality of some of the results presented is not however adequate for publication, as outlined below. Further, a number of errors are evident throughout the paper in regards to axis labels, figure labels and references to figure elements, as outlined below. Lastly, the authors should ensure they have adequately referenced the literature in this area as some relevant findings have not been discussed, as outlined below. They should also discuss the fact that their data pertaining to mechanical regulation of sclerostin expression through activation mTORC1 is only in vitro and requires in vivo validation. To be certain that this is a mechanism, TSC1 KO mice should be loaded to see I the

response to load is suppressed. In vivo other mechanisms may compensate, these cannot be explored in vitro.

We thank the reviewer for acknowledging the significance and interest of our work and the constructive suggestions to help improve our manuscript. And we also thank you for pointing out these issues. We have endeavored to address all these issues in our revised version.

Several literatures have reported that mechanical load can regulate sclerostin expression as well as mTORC1 activity[1, 2]. Loading-induced attenuation of sclerostin expression and elevation of bone formation along with the SBP surface in severe late-stage OA mice.[3] Loading also suppressed Sost expression in adult mice[4], and Sost knockout mice have an enhanced osteogenic response to loading.[5] Our *in vitro* experiments demonstrated that mTORC1 is involved in the mechanical load regulation of sclerostin expression. In order to avoid ambiguity, we decided to delete *in vivo* mechanical load regulation of sclerostin expression related statements in our manuscript.

[1] Jia H, Ma X, Wei Y, *et al.* Loading-Induced Reduction in Sclerostin as a Mechanism of Subchondral Bone Plate Sclerosis in Mouse Knee Joints During Late-Stage Osteoarthritis. *Arthritis & rheumatology* (Hoboken, NJ) 2018; 70(2): 230-41.

[2] Holguin N, Brodt MD, Silva MJ. Activation of Wnt Signaling by Mechanical Loading Is Impaired in the Bone of Old Mice. *Journal of bone and mineral research : the official journal of the American Society for Bone and Mineral Research* 2016; 31(12): 2215-26.

[3] Galea GL, Lanyon LE, Price JS. Sclerostin's role in bone's adaptive response to mechanical loading. *Bone* 2017; 96: 38-44.

[4] You JS, McNally RM, Jacobs BL, *et al.* The role of raptor in the mechanical load-induced regulation of mTOR signaling, protein synthesis, and skeletal muscle hypertrophy. *FASEB journal : official publication of the Federation of American Societies for Experimental Biology* 2019; 33(3): 4021-34.

[5] Marcotte GR, West DW, Baar K. The molecular basis for load-induced skeletal muscle hypertrophy. *Calcified tissue international* 2015; 96(3): 196-210.

Major items:

- 1) Reference should be made to papers discussing HDAC5 (Li 2016) and PTH (Wein 2015) as upstream regulators of sclerostin expression by osteocytes in the introduction or discussion

Thank you for the suggestion. We clarified these points and cited these articles in our revised manuscript. It reads," And the expression of sclerostin was also regulated by HDAC5 and PTH[16, 17]."

[16] Wein MN, Spatz J, Nishimori S, *et al.* HDAC5 controls MEF2C-driven sclerostin expression in

osteocytes. *Journal of bone and mineral research : the official journal of the American Society for Bone and Mineral Research* 2015; 30(3): 400-11.

[17] Li C, Wang W, Xie L, *et al.* Lipoprotein receptor-related protein 6 is required for parathyroid hormone-induced *Sost* suppression. *Annals of the New York Academy of Sciences* 2016; 1364: 62-73.

2) In results section on μ CT data the authors state that the mice show progressive osteopetrosis. This term refers to reduced resorption and given the mechanism for the increased bone mass is not determined by this point in the paper it should be replaced with increased bone mass or volume. Also it states that trabecular separation is increased when it is actually decreased.

Thank you for the valuable comment. We have revised the incorrect information in the text. Now it reads, "Micro-CT analysis of 4-, 8- and 12-week-old mice revealed an increased bone mass in *Tsc1* CKO mice. The distal regions of the femur and the fifth lumbar vertebra exhibited a marked increase in cancellous bone mass in *Tsc1* CKO mice compared with that in DTCL, as reflected in bone mineral density (BMD), trabecular thickness (Tb. Th) and trabecular number (Tb. N), bone volume/tissue volume (BV/TV), as well as a slight decrease in trabecular separation (Tb. Sp) (Figure 1D, Supplementary Figure S2C and D, Supplementary Table S1-3)."

3) The BV/TV data provided in table S1-3 does not denote if it is from femur or vertebra and also the values for BV/TV seem oddly low in the WT mice. Are these values in %?

Thank you for the comment. We apologize for the confusion in table S1-3. We have revised the manuscript and it reads, " Table S1. Micro-CT analysis of femur in DTCL and *Tsc1* CKO mice at 4 weeks of age", " Table S2. Micro-CT analysis of femur in DTCL and *Tsc1* CKO mice at 8 weeks of age", " Table S3. Micro-CT analysis of femur in DTCL and *Tsc1* CKO mice at 12 weeks of age".

The reviewer is correct that BV/TV values on μ CT data are in percentage, and we revised the description from '1' to '%' in our tables.

The reviewer is correct that BV/TV values on μ CT data are in percentage, and we have clarified this confusion. We rechecked all the μ CT data and reperformed our statistic calculations to ensure that all the μ CT data were correct. Then we revised the table from '1' to '%' in our tables and we rewrote our Bv/Tv data in percentage ('19.8 in %').

For further consideration, another likely reason for getting low values in the WT mice might be

the μ CT scanner that we have used. Values between 17% and 21% were reported in the literature [6, 7].

[6] Xu S, Zhang Y, Wang J, *et al.* TSC1 regulates osteoclast podosome organization and bone resorption through mTORC1 and Rac1/Cdc42. *Cell death and differentiation* 2018; 25(9): 1549-66.

[7] Caravaggi P, Liverani E, Leardini A, *et al.* CoCr porous scaffolds manufactured via selective laser melting in orthopedics: Topographical, mechanical, and biological characterization. *Journal of biomedical materials research Part B, Applied biomaterials* 2019.

4) Figure 2F, MAR values are odd, they are unusually high so it is possible the data has not been divided by the days between the labels, 7 days.

Thank you for your careful review and comment. The reviewer is correct. We have recalculated our MAR values. However, the MAR of CKO mice were significantly high. We speculated that the odd speed of bone matrix production was due to those increased dedifferentiation osteocytes. Osteocytes in CKO mice dedifferentiated and regained the activated abilities so they facilitated osteoblasts to produce bone matrix, which caused both high bone mass and odd MAR values.

Figure 2. Deletion of TSC1 in osteocytes stimulates osteogenesis and bone formation in mice.

(F) Mineral apposition rate (MAR) (t test, $p = 0.0062$).

5) Figure S3D the label on the graph does not match how the data is described

Thank you for the comment. We have revised the y-axis in Figure S3D. Now it presents as follow:

Supplementary Figure S3

Figure S3 . Thickness of cortical bone in DTCL and *Tsc1* CKO mice

(A) H&E staining of cortical bone of femora from 4-, 8- and 12-week-old control and *Tsc1* CKO mice. The boxed area was magnified in the panel below. The scale bar represents 100 μm and 50 μm ($n = 6$). (B-C) Histomorphometric measurements showed that (B) the mineralizing surface/bone surface (MS/BS, t test, $p = 0.0087$) and (C) bone formation rate (BFR, t test, $p = 0.0003$) of control mice were lower than that of *Tsc1* CKO mice ($n = 6$). (D) mRNA expression of TSC1 in DTCL and *Tsc1* CKO mice (t test, $p = 0.0007$). Data are represented as mean \pm SD, ** $p < 0.01$, and *** $p < 0.001$.

6) Osteoclast analysis is not conclusive enough. The number of TRAP+ cells should be expressed as a % of bone surfaces, is this the case? It is not clear from the results of methods. Further, both in vivo and in vitro analysis of osteoclasts did not examine their function. This should be examined using serum CTX from mice or examine resorptive capacity on calcium phosphate discs or dentine. Functional analysis is required to truly confirm the impact of TSC1 deletion on bone resorption. The increased osteoclast number may just be a feedback

response given bone formation is up. This should be discussed. Further assessment of serum RANKL and OPG would help determine if the changes in this pathway are only occurring locally to the osteocyte or is it widespread through changes in circulating levels?

Thank you for your comments and valuable suggestions. We agree with the reviewer that the number of TRAP+ cells should be standardized and expressed as a percentage of trabecular surfaces in this case, and we revised Fig 3B. For further osteoclastic functional analysis, we examined osteoclastic resorption capacity *in vitro* using serum CTX from mice and showed in new Fig 2K, and we confirmed that osteoclast function was enhanced in CKO mice.

Figure 2. Deletion of TSC1 in osteocytes stimulates osteogenesis and bone formation in mice. (B) BMD of trabecular bone in the distal femora from 4-, 8- and 12-week-old control (DTCL) and *Tsc1* CKO mice (t test, 4 w: $p = 0.0083$, 8 w: $p = 0.0257$, 12 w: $p = 0.0284$). BMD, bone mineral density (n = 6).

Figure 2. Deletion of TSC1 in osteocytes stimulates osteogenesis and bone formation in mice. (K) ELISA detection of Serum CTX-1 levels in control and *Tsc1* CKO mice (t test, $p = 0.029$, 12-week-old mice, n = 6). All experiments were repeated independently for three times. Data

are represented as mean \pm SD, * $p < 0.05$, ** $p < 0.01$, and *** $p < 0.001$.

We also agree on that the increased osteoclast number may just be a feedback response given bone formation is increased. Our BMD data showed no alteration between CKO and DTCL mice, indicating the quality of bone in CKO mice remained normal. However, μ CT images as well as analysis data showed increased bone mass, which means the quantity of bone in CKO mice increased significantly. In addition, osteocytes are the major source of RANKL. Taken together, we speculated the reason of increased osteoclast number could be a feedback of high bone formation. Meanwhile, increased osteocytes could play an important role in osteoclast number increase.

We have examined both protein and mRNA levels of RANKL and OPG. We think those data were very valuable.

7) A number of protein blots in the paper are not adequate for publication. These include fig 4H and I, 6B and 6D, 7 H and J. These western blots should be repeated and cleaner data presented.

Thank you for the suggestion. We apologize for the low-quality images. We have replaced these figures in our revised manuscript and presented as follows:

Figure 4

Figure 6

Figure 7

8) Figure 6A is difficult to appreciate that the cell morphology changed with loading as there is no image of a non-loaded plate of cells.

Thank you for the comment. In Figure 6A, the 0 h group was non-loaded group, and we clarified these groups in materials and figure legend. Which reads, "The MLO-Y4 osteocyte-like cells were divided into three groups: the control group (no loading to the cultured cells/0 h) and two treatment groups loaded over 12 or 24 h", and "(A) Representative images of the stretch-loaded MLO-Y4 cells for 0, 12 and 24 h. The scale bar represents 100 μm. 0 h: no-loaded group/control group; 12 h: stretch-loaded for 12 h; 24 h: stretch-loaded for 24 h."

Figure 6

Figure 6. Mechanical loading activates mTORC1 to prevent sclerostin expression in osteocytes.

(A) Representative images of the stretch-loaded MLO-Y4 cells for 0, 12 and 24 h. The scale bar represents 100 μm . 0 h: no-loaded group/control group; 12 h: stretch-loaded for 12 h; 24 h: stretch-loaded for 24 h.

9) Page 13 reference to rapamycin rescuing the stretch-loading inhibition of sclerostin is incorrect based on the data in figure 6B. Unless the reviewer is interpreting the figure incorrectly, SOST is still significantly reduced with rapamycin treatment. The fact that in all of the figure legends fail to confirm the comparison being made for each ** symbol does not help with this confusion.

Thank you for the suggestion. We have clarified this confusion in the revised manuscript. It reads, "Furthermore, rapamycin could slightly affect stretch-loading inhibited sclerostin expression (Figure 6D and E)." Each ** symbol means P value less than 0.01 and P value are presented in corresponding figure legends.

Figure 6

Figure 6. Mechanical loading activates mTORC1 to prevent sclerostin expression in osteocytes.

(A) Representative images of the stretch-loaded MLO-Y4 cells for 0, 12 and 24 h. The scale bar represents 100 μ m. 0 h: no-loaded group/control group; 12 h: stretch-loaded for 12 h; 24 h: stretch-loaded for 24 h. (B) Western blot analysis of SOST, pS6 and pS6K expression in MLO-Y4 cells stretch-loaded for 0, 12 or 24 h. (C) mRNA levels of SOST in stretch-loaded MLO-Y4 cells (one-way ANOVA with Bonferroni multiple comparison, 0 h vs 12 h, $p = 0.0077$; 0 h vs 24 h, $p = 0.0003$, $n = 6$). (D, E) MLO-Y4 cells were stretch-loaded and treated with 1 nM rapamycin for 24 h, then cell lysates were subjected to (D) western blotting or (E) qPCR analysis for SOST (non-parametric statistical test, vehicle group, $p = 0.0246$; ΔR group, $p = 0.0473$, $n = 6$). All

experiments were repeated independently for three times. NL, no-loaded; SL, stretch-loaded. Data are represented as mean \pm SD, * $p < 0.05$, ** $p < 0.01$, and *** $p < 0.001$.

10) Reference to figure 7J and K, Sirt1 siRNA did not appear to rescue scl expression blocked by TSC1 shRNA, but this is the interpretation by authors. Again the ** reference is not clear and may make this clearer to the reader.

Thank you for your great suggestion. We agree with the reviewer that Sirt1 siRNA did not appear to rescue scl expression blocked by TSC1 shRNA. Our data showed sirt1 could inhibit sclerostin expression which was presented in Figure 7J and K. Each ** symbol means P value less than 0.01 and P values are presented in corresponding figure legends. We clarified these data in our revised manuscript. It presents as follow:

Figure 7

Figure 7. TSC1 promotes sclerostin expression in osteocytes partially through Sirt1.

(A) Bone lysates from 10-week-old control and *Tsc1* CKO mice were subjected to western blot analysis for Sirt1. (B) mRNA expression of Sirt1 in femora of 10-week-old control (DTCL) and *Tsc1* CKO mice (t test, $p = 0.0005$, $n = 6$). (C) Western blot analysis of Sirt1 expression in MLO-Y4 cells infected with TSC1 shRNA lentivirus. (D) mRNA levels of Sirt1 in MLO-Y4 cells infected with NC and TSC1 shRNA lentivirus (t test, $p = 0.0009$, $n = 6$). (E) Western blot analysis of Sirt1 in MLO-Y4 cells treated with vehicle (V) and 1 nM of rapamycin for 48 h (Δ R). (F) mRNA levels of Sirt1 in MLO-Y4 cells treated with 1 nM rapamycin for 48 h (Δ R) (t test, $p = 0.0008$, $n = 6$). (G) MLO-Y4 cells were transfected with Sirt1 or negative control (NC) siRNA, (H) then treated with 1 nM

rapamycin (ΔR) for 48 h, and cell lysates were subjected to western blot analysis for SOST. (I) mRNA level of SOST was assayed by qPCR (non-parametric statistical test, NC group, $p = 0.0027$; Sirt1 siRNA group, $p = 0.0402$). (J) MLO-Y4 cells infected with TSC1 shRNA lentivirus ($\Delta TSC1$) for 72 h, subsequently transfected with Sirt1 or negative control siRNA for another 48 h, then SOST expression was detected by western blotting. (K) mRNA level of SOST was assayed by qPCR (non-parametric statistical test, NC group, $p = 0.0033$; Sirt1 siRNA group, $p = 0.0021$). All experiments were repeated independently for three times. Data are represented as mean \pm SD, $*p < 0.05$, $**p < 0.01$, and $***p < 0.001$.

11) All in vivo data presented is showing male mice. Did female mice show a similar phenotype? This should be discussed as sex can manifest its own impact on the skeleton.

Thank you for your valuable comment. As the reviewer mentioned in this comment, sex does affect skeletal development all along their growth in mammals. In such circumstances, experimental objects are required in same gender for most skeletal development studies due to the impact of estrogen on bone development. Therefore, we used all male mice as experimental objects for in vivo data in case of the interference of estrogen. We have clarified the manuscript on page 15. Now it reads, "Sex does affect skeletal development all along their growth in mammals. Therefore, we used all male mice as experimental objects for in vivo data in case of the interference of estrogen. Male experimental and control littermate mice were compared at 4, 8 and 12 weeks of age."

Minor items:

Page 5 first paragraph, monoclonal antibodies to sclerostin inhibit its activity but do not reduce sclerostin. This should be clarified/ Also

Thank you for your comment. We have clarified the text on page 5. Now it reads, "Treatment with anti-sclerostin monoclonal antibodies inhibit the activity of sclerostin, thus improving bone mass and bone strength, along with enhancing repair of fractures in mice and rats[23, 24]."

[23] Li X, Zhang Y, Kang H, *et al.* Sclerostin binds to LRP5/6 and antagonizes canonical Wnt signaling. *The Journal of biological chemistry* 2005; 280(20): 19883-7.

[24] Niziolek PJ, Farmer TL, Cui Y, *et al.* High-bone-mass-producing mutations in the Wnt signaling pathway result in distinct skeletal phenotypes. *Bone* 2011; 49(5): 1010-9.

Page 7, paragraph 1 last line “Altogether, these data suggest that TSC1 deletion and mTORC1 activation in osteocyte markedly increase bone mass in mice. Should read Altogether, these data suggest that TSC1 deletion and mTORC1 activation in osteocytes markedly increase bone mass in mice.

Thank you for the comment. We have revised our expression to “Altogether, these data suggest that TSC1 deletion and mTORC1 activation in osteocytes markedly increased bone mass in mice” in our manuscript.

Page 9 line 3 showed a dramatic increase of RANKL....this change is not that obvious so dramatic feels like and over statement here.

Thank you for your great suggestion. We have revised the text in our manuscript. Now it reads, “Western blot analysis of bone lysates showed increased RANKL in *Tsc1* CKO mice (Figure 3C).”

Figures 5I and 5L are not referred to anywhere in the text of the paper, this should be corrected

Page 14 line 2, reference to figure 7G and I should be 7G and H.

Thank you for the comment. We apologize for the missing information and we have revised the text. Now it reads, “We found that anti-sclerostin antibody rescued MLO-Y4 Δ Rap medium-inhibited MC3T3-E1 osteogenesis (Figure 5H, I and J), while recombinant sclerostin blocked MLO-Y4 Δ TSC1 medium-stimulated MC3T3-E1 osteogenesis (Figure 5K, L and M)”. “ We found that the Sirt1 siRNA significantly increased the level of sclerostin (Figure 7G, H and I)”

Referee: 2

Comments to the Author(s)

The manuscript by Liu et al examines the role of conditional deletion of tuberous sclerosis complex 1 (TSC1) in the skeletal homeostasis. Liu et al show that in growing mice, conditional deletion of TSC1 results in a high bone remodeling phenotype with a positive balance that leads to bone gain. In addition, they found that mice lacking TSC1 in osteocytes have decreased expression of Sost/Sclerostin and increase expression of Rankl. Mechanistic studies showed that TSC1 activates mTORC1 signaling, which in turn activates Sirt to downregulate Sost/Scl expression and favor osteoblast differentiation. Further, authors found that mTORC1 signaling is also

involved in the regulation of Sost by mechanical stimulation using in vitro approaches. This work presents a remarkable bone phenotype induced by deletion of TSC1 in osteocytes. The studies are technically very well done and the work is likely to be of significant interest to a wide range of researchers. However, here are some issues that need to be modified before the manuscript can be considered for publication.

Thank you for acknowledging the interest of our work and we sincerely appreciate your valuable comments.

1. "In summary, using a genetic approach, we clarified the role of TSC1/mTORC1 in controlling sclerostin secretion and bone formation (Figure 9)" in this part should be changed to Figure 8.

Thank you for the comment. We have revised the incorrect information in our revised manuscript. Now it reads, "In summary, using a genetic approach, we clarified the role of TSC1/mTORC1 in controlling sclerostin secretion and bone formation (Figure 8)"

2. In Figure legend part, H should be changed to G in Figure 1.

Thank you for the comment. We have revised the incorrect information in our revised manuscript. Now it reads," (H) H&E staining of long bone from 4-, 8-, and 12-week-old Tsc1 CKO and control mice. The scale bars represent 100 μ m (n = 6)"

3. In Figure 5 J, vertical ordinate expression should be changed to relative expression of OCN.

Thank you for the comment. We have revised the incorrect information in our revised manuscript.

4. In Figure 5 M, vertical ordinate expression should be changed to relative expression of OCN.

Thank you for the comment. We have revised the incorrect information in our revised manuscript.

5. The study claims that osteoclastogenesis and osteogenesis are both enhanced in Tsc1CKO mice. Please clarify the mechanism of osteoclast activities and its affect on bone formation.

Thank you for your valuable comment. We have demonstrated osteocytes are a major source of RANKL to stimulate osteoclastogenesis during bone remodeling [47, 48] on page 13. We speculate the increased osteogenesis was due to sclerostin negatively regulating β -catenin expression in osteoblasts, and the increased osteoclastogenesis was owing to the enhanced RANKL expression in osteocytes.

47. Nakashima, T., Hayashi, M., Fukunaga, T., Kurata, K., Oh-Hora, M., Feng, J. Q., Bonewald, L. F., Kodama, T., Wutz, A., Wagner, E. F., Penninger, J. M. & Takayanagi, H. (2011) Evidence for osteocyte regulation of bone homeostasis through RANKL expression, *Nature medicine*. **17**, 1231-4.

48. Xiong, J., Onal, M., Jilka, R. L., Weinstein, R. S., Manolagas, S. C. & O'Brien, C. A. (2011) Matrix-embedded cells control osteoclast formation, *Nature medicine*. **17**, 1235-41.

We would like to express our appreciation to you and the reviewers for your comments to help improve our manuscript.

Yours sincerely,

Xiaochun